ⓐ | **Open Peer Review** | Clinical Microbiology | Research Article

# Characterization of a novel *Phietavirus* genus bacteriophage and its potential for efficient transfer of modified shuttle plasmids to *Staphylococcus aureus* strains of different clonal complexes

Łukasz Kałuski,[1,2] Emil Stefańczyk,[1] Aleksandra Głowacka-Rutkowska,[1] Jan Gawor,[1] Joanna Empel,[3] Monika Orczykowska-Kotyna,[3] Aleksandra Szczypkowska,[1] Karolina Żuchniewicz,[1] Robert Gromadka,[1] Małgorzata Łobocka[1]

**ABSTRACT** *Staphylococcus aureus* is a significant human pathogen responsible for various nosocomial and community-acquired infections, leading to considerable morbidity and mortality worldwide. Temperate bacteriophages contribute to its virulence and facilitate the dissemination of pathogenicity traits. We isolated a novel siphovirus of the *Phietavirus* genus, ASZ22RN, derived from a prophage of an *S. aureus* clonal complex 7 strain and capable of propagating in the prophage-free laboratory strain RN4220. ASZ22RN either productively infected or lysed from without all 47 tested *S. aureus* clinical strains across 12 clonal complexes (CCs), demonstrating its ability to puncture their cell envelopes. When ASZ22RN was propagated in RN4220 cells harboring an *S. aureus-Escherichia coli* plasmid replicating via theta mode, it transduced the plasmid to plasmid-free RN4220 with low frequency. The transduction frequency increased by nearly five orders of magnitude when the plasmid contained a fragment of ASZ22RN DNA (*terS*). Most *terS+* plasmid-transducing particles carried plasmid concatamers, while some carried plasmid-phage DNA hybrids, as demonstrated by DNA sequencing. Strains from all tested CCs served as recipients for transduction, regardless of the presence of type I restriction-modification enzymes targeting plasmid/phage DNA, or prophages with lysis-lysogeny switch regions conferring superinfection immunity to ASZ22RN. Our results indicate that intracellular phage defense systems do not prevent phage-mediated plasmid transfer and demonstrate a simple method for introducing plasmids constructed in *E. coli* into clinical *S. aureus* isolates. Moreover, the presence of the ASZ22RN lysis-lysogeny switch region in 21% of tested ASZ22RN-resistant strains highlights superinfection exclusion as a dominant mechanism of resistance to siphoviruses in staphylococci.

**IMPORTANCE** This study highlights the capacity of a newly isolated staphylococcal *Phietavirus*, ASZ22RN, to transfer a low-copy-number shuttle *Staphylococcus aureus-Escherichia coli* plasmid to various *S. aureus* strains representing major clonal complexes from among clinical isolates. By increasing the plasmid transduction efficiency in an ASZ22RN-specific manner, we show that the primary factor determining a given strain's ability to be a recipient in transduction is the capacity of transducing phage to puncture the cell envelopes of this strain. This can be determined not only based on productive phage infection but also lysis from without. Major intracellular mechanisms protecting *S. aureus* from productive phage infection do not impede the transduction-mediated acquisition of plasmids. Moreover, the lack of phage DNA in most of the plasmid-transducing virions indicates the lack of phage contamination in most transductants. Our results offer a promising approach for developing efficient pipelines to introduce plasmids constructed in *E. coli* to clinical *S. aureus* isolates.

Address correspondence to Małgorzata Łobocka, lobocka@ibb.waw.pl.

The authors declare no conflict of interest.

See the funding table on p. 27.

**KEYWORDS** *Staphylococcus aureus*, temperate bacteriophage, plasmid, transduction, horizontal gene transfer, *Phietavirus*, superinfection exclusion

*S*taphylococcus aureus is one of the most extensively studied bacterial pathogens. Its pathogenicity is significantly influenced by various prophages (1). Most tested *S. aureus* strains are lysogens and can carry up to five different prophages (2). Certain temperate *S. aureus* phages have been identified as carriers of pathogenicity-associated traits. They not only spread among staphylococcal strains by themselves but also contribute to the dissemination of staphylococcal pathogenicity islands, plasmids, *SCCmec* cassettes, and other mobile genetic elements (3–6). The aforementioned properties also make them potential vesicles for transferring the desired DNA between staphylococcal cells in studies on this bacterium.

The most common staphylococcal temperate phages have siphovirus morphology and genome size within the range of 39–47 kB (2, 7). They were assigned to one cluster (designated as B) based on overall similarities (7). They represent five genera of two subfamilies: *Phietavirus*, *Dubowvirus,* and *Triavirus* genera of the *Azeredovirinae* subfamily, and *Peeveelvirus* and *Biseptimavirus* genera of the *Brofenbrennervirinae* subfamily (8). Their genomes are composed of a few functionally conserved modules associated with DNA packaging and virion morphogenesis, cell lysis, lysogeny control, DNA replication and recombination, and transcriptional regulation (7). Differences in functionally conserved genes that define these modules form the basis of the cluster B phages grouping scheme proposed by Kahánková et al. (9). It utilizes multiplex PCR detection of various types of integrase (*int*), antirepressor, replication protein (*polA*, *dnaC*, and *dnaD*), dUTPase, portal protein, tail appendices, and endolysin genes to include each phage into a particular group or subgroup. Significant similarities of certain modules or their genes between phages representing different genera are a hallmark of frequent gene exchanges within the entire group of these phages.

Cluster B phages utilize at least eight different attachment sites in the *S. aureus* genome to integrate with the host DNA (2). Three of them were mapped to protein coding genes *geh, hlb,* and locus SAV1877 (*yfkA*) of unknown function (10). The interruption of *geh* and *hlb* genes by a prophage is associated with the loss of glycerol ester hydrolase and beta-hemolysin function, respectively, and the relevant phenotype change (2, 11, 12).

Lysogeny is associated with the inhibition of lytic development at the level of transcriptional control of early genes and with the expression of certain phage-encoded functions adaptive for a host (2, 13). The control is exerted by the lysis-lysogeny control module, which is organized analogously to the one of phage lambda. It contains a regulatory DNA region (switch region) that can bind repressors encoded by oppositely oriented flanking genes analogous to lambda CI and Cro (14, 15). It also contains the integrase-encoding genes and the phage DNA region (*attP*) interacting with the bacterial attachment site for a phage (*attB*) in a host chromosome (16). The region between the integrase-encoding gene of the lysis-lysogeny control module and the endolysin gene of the cell lysis module in cluster B phage genomes corresponds to the left or the right end of their prophage DNA. It is typically occupied by accessory genes. Some of them encode bacterial adaptive factors, such as toxins, innate immunity modulators, or other virulence determinants increasing the pathogenicity of their lysogens (2, 7, 10, 17, 18). Accessory genes are strongly associated with phages from specific *int* groups (9). For instance, genes encoding exfoliative toxin A (*eta*) or Panton-Valentine leukocidin (*lukS-PV* and *lukF-PV*) were detected predominantly in phages of integrase type 1 (Sa1) and 2 (Sa2), respectively. The so-called immune escape complex, which includes up to five virulence factors, is encoded predominantly by phages of integrase type 3 (Sa3) (13). Virulence-associated factors of integrase type 4, 6, 8, 9, and 12 phages, if any, still await identification (2).

Based on the packaging specificity, staphylococcal cluster B siphoviruses can be divided into two groups. Phages of *Triavirus, Peeveelvirus,* and *Biseptimavirus* genera

pack their DNA from genome concatamers (products of phage DNA replication) based on the sequence-specific recognition of genome unit ends (*cos* sites) by the phage terminase complex. The complex introduces a specific cut at each *cos* at packaging, initiating and terminating each head filling with a *cos*-terminated genome monomer (6). The *cos* phages could potentially transfer DNA by specialized transduction, analogous to that mediated by phage Λ (19). It is a rare event associated with imprecise excision of prophage DNA at induction of lytic development and results in the transfer of host DNA fragments from the border between prophage and host DNA. Recently, the phages dependent on the *cos* sites for packaging were also demonstrated to transfer DNA of specific staphylococcal pathogenicity islands (20, 21).

Phages belonging to the *Phietavirus* and *Dubowvirus* genera pack their DNA by a headful mechanism (18, 22, 23). The packaging is initiated from the first *pac* site in each genome concatamer—the product of phage replication. It ends with the unspecific cut when the head is full and starts again from this imprecise cut to fill in the next head (reviewed by reference 24). As a result, DNA molecules in virions are terminally redundant and cyclically permuted. So-called pseudo-*pac* sites similar in sequence to *pac* and occasionally present in a chromosome can also initiate packaging, leading to the formation of phage particles transferring chromosomal DNA by generalized transduction. In turn, in the process of so-called lateral transduction, packaging is initiated from the *pac* site before prophage excision from a chromosome (21, 25). It proceeds further through the headful mechanism leading to the formation of phage particles containing chromosomal fragments up to several hundred kilobases downstream of the integrated prophage. Packaging of staphylococcal pathogenicity islands (SaPIs), which parasitize certain phages, depends on SaPI-mediated redirection of phage development toward the production of modified virions of smaller phage heads (26). They can accommodate SaPI DNA, but not phage DNA. Several SaPI-encoded proteins participate in regulatory processes associated with SaPI excision, small-headed virion production, and packaging. Those directed to the reprogramming of phage development target essential phage proteins.

The genomic sequences of 56 *Phietavirus* and 46 *Dubowvirus* genus phages have been deposited in GenBank (accessed 8 May 2024). Some of these phages have been studied in detail with respect to their interaction with SaPIs, expression of pathogenicity-associated traits, or the ability to transfer genetic material by transduction (2, 6). Although some properties of their representatives have been described, knowledge about their coding potential has several gaps, and the use of these phages as genetic tools has been limited to a few specialized laboratories. Here, we describe a new temperate *S. aureus* phage, analyze its genomic and physiological properties, and provide the analysis of its transducing potential with an emphasis on its use in plasmid transfer from laboratory to clinical *S. aureus* strains. We also show for the first time how significant the role of superinfection immunity is in limiting the spread of siphoviruses in the *S. aureus* population.

## RESULTS

### Phage isolation, morphology and physiological parameters

Bacteriophage vB_SauS_ASZ22RN (ASZ22RN) was isolated from a mixed culture of 14 *S. aureus* strains (Table S1, strains marked with #) as infecting laboratory strain RN4220. It formed plaques with an average diameter of~1.3 mm on a layer of RN4220 strain cells (Fig. 1A). Virions of ASZ22RN appeared to have siphovirus morphology with icosahedral head and a long, flexible tail terminated with a mace-shaped baseplate. Average head diameters and tail lengths were 57 and 170 nm, respectively, including the 28 nm baseplate (Fig. 1B and C; Table S2). The ASZ22RN virions could form aggregates through the interaction of their baseplates.

The phage could be efficiently propagated in cells of prophage-free RN4220 strain, with a 35 min latent period and burst size of ~13 virions per cell (Fig. S1). However, growing colonies of resistant cells were observed on plaques upon longer incubation (3–

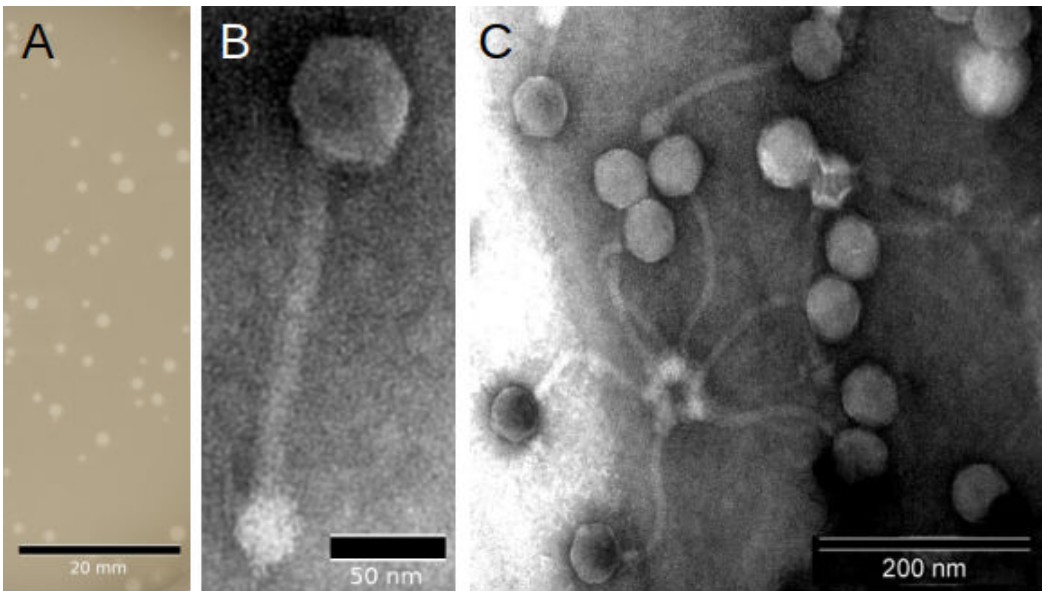

**FIG 1** Morphology of plaques formed by ASZ22RN on a layer of *S. aureus* RN4220 strain cells (A) and transmission electron micrographs of ASZ22RN virions stained with uranyl acetate (B, C).

4 days). Some of them represented lysogens as determined by the formation of amplicons in colony PCR with their DNA as a template and primers specific for phage ASZ22RN DNA (Fig. S2A). The lysogens were resistant to reinfection with ASZ22RN as indicated by the lack of plaques in a spot test with lysogens used as indicators and serially diluted suspensions of ASZ22RN. Only with the highly concentrated phage suspension ($10^{10}$–$10^{11}$ PFU/mL) one could see a turbid lysis zone (likely representing lysis from without) in place of the phage suspension drops (Fig. S2B). Lytic development of ASZ22RN in lysogens could be induced by the induction of SOS response upon treatment with mitomycin C. Cells of lysogen cultures incubated with mitomycin C lysed. The phage titer of lysates after 3 h or overnight incubation with mitomycin C and filtering were $10^9$ and $10^7$, respectively. We attribute the drop in phage titer in overnight cultures of lysed cells as compared to 3 h cultures to phage adsorption to the remnants of lysed cells.

## Bacteriophage ASZ22RN genome, phylogeny, and DNA packaging strategy

The sequence reads of the ASZ22RN genome assembled into a circular contig of 43,579 bp with no distinct regions of significantly increased coverage, which is indicative of phages that pack their DNA by a headful mechanism (27). The assembled genomic sequence appeared to be highly similar to the genomic sequences of *Phietavirus* genus phages of *Caudoviricetes* class (Fig. 2A and B). The closest relative of ASZ22RN is *Phietavirus* 3MRA (NC_028917.1), which shares 93.9% genomic sequence with ASZ22RN. Based on these criteria, the ASZ22RN phage can be classified to the genus *Phietavirus*. The identity of the ASZ22RN genomic sequence to its closest relatives from this genus is below the demarcation criteria for phages representing the same species (<95%) (28), indicating that ASZ22RN represents a new species of the *Phietavirus* genus. We propose to designate it *Phietavirus ASZ22RN*. Although phage ASZ22RN shares 93.9% genomic sequence with that of 3MRA, its sequence is only 72%, 71.4%, and 71.3% identical to the sequences of the next most closely related *Phietavirus* phages, vB_SauS_I73, phiJB, and 96, respectively. In the phylogenetic tree of related phietaviruses, ASZ22RN groups in the same branch as phages B166, B122, and phiETA3, which encode exfoliative toxin A ([17]; GenBank acc. no. AP008954) (Fig. 2B). *In silico* analysis of the ASZ22RN genome with primers of the Kahánková grouping scheme (9) positioned this phage among staphylococcal siphoviruses of integrase group 5 (Sa5int), antirepressor group 1a (ant1a),

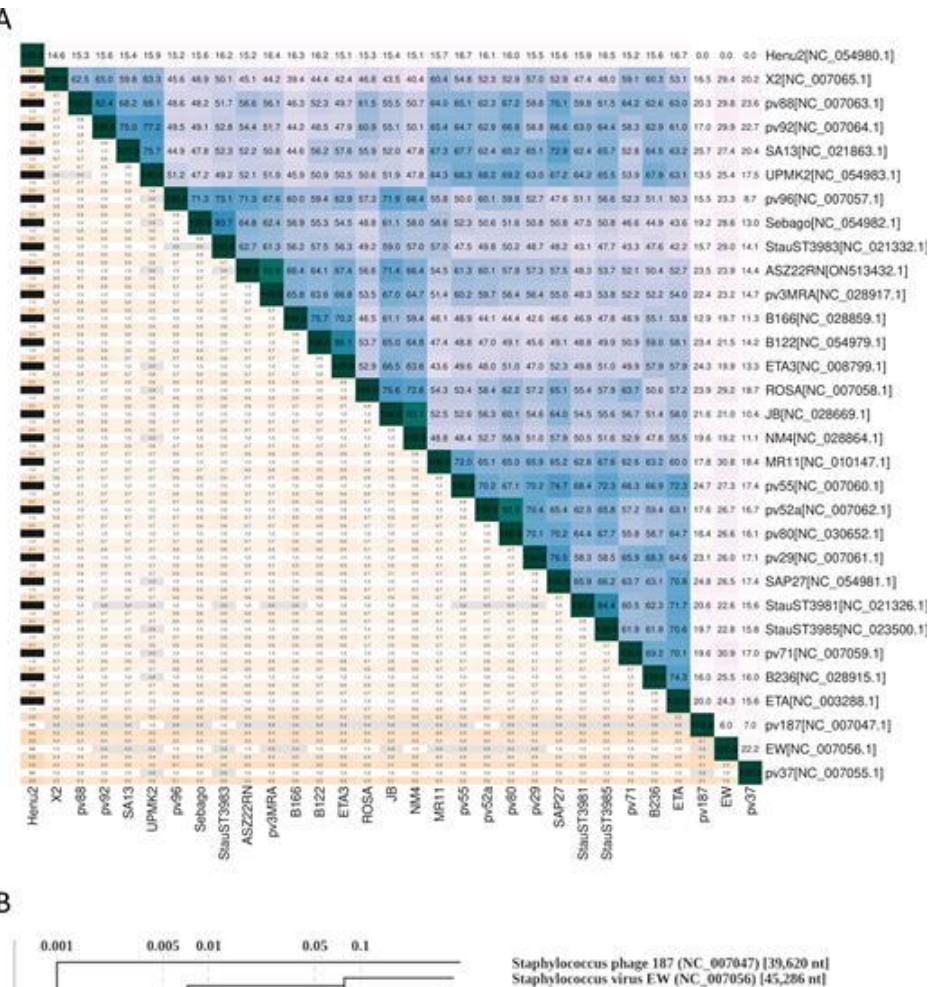

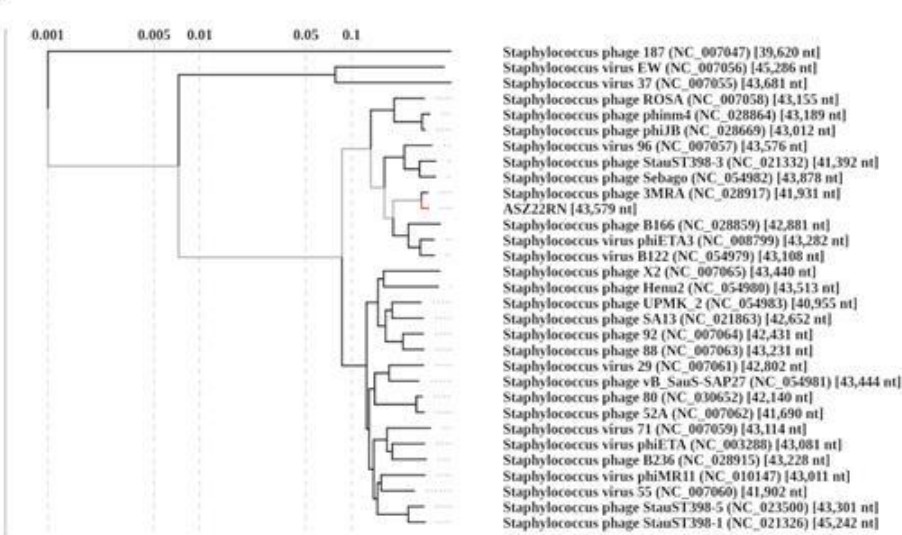

**FIG 2** Taxonomic assignment of ASZ22RN bacteriophage. (A) Pairwise intergenomic similarities between genomes of phietaviruses calculated using Virus Intergenomic Distance Calculator with default parameters. Genome similarities (above the diagonal line) are indicated with color intensity, from white (no relation), through blue (somewhat similar) to dark green (identical). The aligned fractions for each genome (below the diagonal line) are marked using three indicator values for each compared pair from top to bottom: aligned fraction genome 1, genome length ratio, and aligned fraction genome 2. Orange-to-white and black-to-white color gradients emphasize alignments and genome length ratio, respectively, thus whiter colors correspond to genome pairs with higher similarity values. (B) Phylogenetic analysis of ASZ22RN and the most closely related phages based on whole-genome-wide sequence similarities calculated by tBLASTx with the use of ViPTree. The ASZ22RN phage is marked with red.

**TABLE 1** Bacteriophage ASZ22RN genes and their predicted products

| Gene no. | Gene position (bp) | Strand[e] | Length of product (aa) | Predicted molecular mass (kDa) | pI | Known or predicted function | Protein ID | The closest homolog in GenBank (GenBank acc. no.) | Coverage[d] (%) | Identity[d] (%) | Amino acid sequence motifs |
|---|---|---|---|---|---|---|---|---|---|---|---|
| 1 | 1–495 | + | 164 | 18.72 | 5.45 | Terminase small subunit | WCS65153.1 | DW2 YP_009045030.1 | 100 | 99.39 | Coil 122–142[a] Terminase_2 (4–102)[c] |
| 2 | 488–1,711 | + | 407 | 47.15 | 6.10 | Terminase large subunit | WCS65154.1 | 3MRA YP_009209267.1 | 100 | 100 | Terminase-6N (30–237)[c] |
| 3 | 1,708–3,132 | + | 474 | 55.04 | 4.69 | Portal (connector) protein | WCS65155.1 | ETA3 YP_001004371.1 | 100 | 100 | Phage_prot_Gp6 (20–454)[c] |
| 4 | 3,101–4,054 | + | 317 | 36.79 | 9.03 | Minor capsid protein, putative ejection protein | WCS65156.1 | IME1364_02 QPN96421.1 | 100 | 100 | PhageSPP1_gp7 (157–275) Phage_Mu_F (152–276)[c] |
| 5 | 4,056–4,262 | + | 68 | 7.83 | 4.51 | Scaffold protein | WCS65157.1 | phiJB YP_009188718.1 | 100 | 100 | Coil (14–35)[a] |
| 6 | 4,365–4,949 | + | 194 | 22.68 | 5.15 | Scaffold protein | WCS65158.1 | phiJB YP_009188719.1 | 100 | 100 | DUF4355 domain (43–162)[c] |
| 7 | 4,966–5,880 | + | 304 | 33.60 | 5.05 | Major capsid protein | WCS65159.1 | ETA3 YP_001004375.1 | 100 | 100 | Phage_capsid (18–301)[c] |
| 8 | 5,889–6,035 | + | 48 | 5.52 | 9.35 | | WCS65160.1 | ETA3 YP_001004376.1 | 100 | 100 | – |
| 9 | 6,041–6,391 | + | 116 | 13.67 | 7.81 | Head-tail adaptor protein | WCS65161.1 | vB_SauS_287 UKM35846.1 | 100 | 100 | SPP1 gp15 (6–108)[a] |
| 10 | 6,403–6,738 | + | 111 | 12.94 | 5.56 | Head completion protein | WCS65162.1 | B166 YP_009204072.1 | 100 | 100 | Phage_SPP1_heat-tail_adaptor (28–102)[a] |
| 11 | 6,725–7,138 | + | 137 | 15.16 | 4.98 | HK97 gp10 family protein | WCS65163.1 | 96 YP_240238.1 | 100 | 100 | HK97-gp10_like (9–82)[c] |
| 12 | 7,151–7,576 | + | 141 | 16.41 | 5.59 | Tail completion protein | WCS65164.1 | ETA3 YP_001004380.1 | 100 | 100 | GP17_like (13–126)[a] DUF 3168 (12–127)[c] |
| 13 | 7,577–8,134 | + | 185 | 20.52 | 4.81 | Major tail protein, tail tube protein | WCS65165.1 | ETA3 YP_001004381.1 | 100 | 100 | – |
| 14 | 8,201–8,707 | + | 168 | 19.30 | 5.02 | Putative phage tail assembly chaperone | WCS65166.1 | IME1364_02 QPN96369.1 | 100 | 100 | Phage_TAC_12 (9–124)[a] |
| 15 | 8,752–9,036 | + | 94 | 11.14 | 9.81 | | WCS65167.1 | phiJB YP_009188728.1 | 100 | 100 | – |
| 16 | 9,040–12,183 | + | 1,047 | 113.51 | 9.27 | Tail tape-measure protein | WCS65168.1 | 96 YP_240243.1 | 100 | 100 | Coil (6–26, 71–98) TMhelix (229–251, 266–288, 569–591, 665–687, 731–753) Apolipoprotein (796–908) FELS-2 prophage protein (826–954)[a] Transmembrane (1)[b] |

*(Continued on next page)*

TABLE 1 Bacteriophage ASZ22RN genes and their predicted products (Continued)

| Gene no. | Gene position (bp) | Strand[e] | Length of product (aa) | Predicted molecular mass (kDa) | pI | Known or predicted function | Protein ID | The closest homolog in GenBank (GenBank acc. no.) | Coverage[d] (%) | Identity[d] (%) | Amino acid sequence motifs |
|---|---|---|---|---|---|---|---|---|---|---|---|
| 17 | 12,198–13,139 | + | 313 | 36.32 | 5.48 | Distal tail protein | WCS65169.1 | 96 YP_001551741.1 | 100 | 100 | Sipho_tail (31–313)[c] |
| 18 | 13,150–15,036 | + | 628 | 70.54 | 7.23 | Tail-associated lysin (Tal) | WCS65170.1 | phiJB YP_009188731.1 | 100 | 100 | Coil (373-393) SGNH_hydrolase (411-621)[a] Prophage_tail (90-331)[c] (421-613) Lipase_GDSL_2 (420-614) |
| 19 | 15,049–16,947 | + | 632 | 73.01 | 7.30 | Receptor-binding protein (RBP) | WCS65171.1 | 96 YP_240249.1 | 100 | 99.68 | P68_RBP_TagC-like_b_propeller (159–416)[a] |
| 20 | 16,947–18,770 | + | 607 | 67.01 | 4.80 | FibL (lower tail fiber) | WCS65172.1 | phiJB YP_009188733.1 | 100 | 99.67 | Coil 148–168[a]; BppU_N (9-160)[c] |
| 21 | 18,770–19,147 | + | 125 | 14.19 | 4.54 | Lower fiber, putative receptor-binding protein | WCS65173.1 | phiJB YP_009188734.1 | 100 | 99.20 | Coil (97-117)[a] DUF2977 (1–61)[c] |
| 22 | 19,148–19,324 | + | 58 | 7.02 | 5.28 | putative chaperone | WCS65174.1 | MR11 YP_001604150.1 | 100 | 100 | XkdX (4–48)[a] |
| 23 | 19,365–19,664 | + | 99 | 11.83 | 9.57 | SLT orf99 homolog | WCS65175.1 | MR11 YP_001604151.1 | 100 | 100 | DUF2951 (2–99) TMhelix (76–98)[a] Transmembrane (1)[b] |
| 24 | 19,801–21,675 | + | 624 | 71.08 | 9.52 | Tail tip cell wall peptidoglycan hydrolase | WCS65176.1 | 29 YP_240556.1 | 100 | 99.04 | Glucosaminidase (485-613) CHAP (29-119)[c] |
| 25 | 21,688–22,926 | + | 412 | 46.13 | 5.24 | Upper tail fiber | WCS65177.1 | vB_SauS_I73 UKM36462.1 | 100 | 98.54 | Coil 244–264[a] Bpu_N (3-146); PTR (316-359)[c] |
| 26 | 22,931–23,326 | + | 131 | 14.43 | 8.75 | | WCS65178.1 | 3MRA YP_009209244.1 | 100 | 100 | – |
| 27 | 23,382–23,819 | + | 145 | 15.66 | 4.84 | Holin | WCS65179.1 | MR11 YP_001604155.1 | 100 | 100 | TMhelix (13–35, 39–61) Signal peptide (1–26)[a] Transmembrane (2)[b] phage_holin_1 (3–82)[c] |
| 28 | 23,800–25,245 | + | 481 | 54.03 | 8.62 | N-acetylmuramoyl-L-alanine amidase | WCS65180.1 | 3MRA YP_009209309.1 | 100 | 99.79 | CHAP (20-113); Amidase_2 (196-324); SH3_5 (395-460)[a] |
| 29 | 25,521–25,420 | - | 33 | 3.91 | 9.51 | | WVM05242.1 | phage 96 YP_240260.1 | 100 | 100 | – |
| 30 | 25,793–25,623 | - | 56 | 6.59 | 8.66 | | WVM05243.1 | 3MRA NC_028917.1 (39722–39889) | 100 | 100 | – |
| 31 | 25,989–25,867 | - | 40 | 4.75 | 10.46 | Putative membrane protein | WVM05244.1 | vB_SauS_I73 | 100 | 100 | TMhelix (7–29) |

**TABLE 1** Bacteriophage ASZ22RN genes and their predicted products (Continued)

| Gene no. | Gene position (bp) | Strand[e] | Length of product (aa) | Predicted molecular mass (kDa) | pI | Known or predicted function | Protein ID | The closest homolog in GenBank (GenBank acc. no.) | Coverage[d] (%) | Identity[d] (%) | Amino acid sequence motifs |
|---|---|---|---|---|---|---|---|---|---|---|---|
| | | | | | | | | OM439672.1 (43434–43315) | | | Signal peptide (1–30)[a] |
| 32 | 26,131–27,177 | - | 348 | 41.32 | 9.55 | integrase | WCS65181.1 | MR25 (Dubowvirus) YP_001949799.1 | 100 | 100 | Phage_integrase (163-338), Arm-DNA-bind_4 (11–55), Phage_int_SAM_3 (60-114)[c] |
| 33 | 27,247–27,717 | - | 156 | 18.10 | 4.99 | Putative membrane protein | WCS65182.1 | B166 YP_009204029.1 | 100 | 100 | Coil (57-99, 114-134), Tmhelix (135-155)[a], Transmembrane (1)[b] |
| 34 | 27,710–28,231 | - | 173 | 20.59 | 5.22 | | WCS65183.1 | 3MRA YP_009209306.1 | 100 | 100 | – |
| 35 | 28,263–28,397 | - | 44 | 5.39 | 9.70 | Putative membrane protein | WCS65184.1 | B166 YP_009204031.1 | 100 | 100 | TMhelix (15–37)[a] |
| 36 | 28,415–28,876 | - | 153 | 18.03 | 4.96 | Putative Zn-dependent metalloprotease[H] | WCS65185.1 | 3MRA YP_009209304.1 | 100 | 100 | – |
| 37 | 28,889–29,212 | - | 107 | 12.43 | 8.89 | C1-like repressor | WCS65186.1 | StauST398-5 YP_009002826.1 | 100 | 100 | Cro/CI-type_HTH (4–78), lambda_DNA_bd_dom_sf (4–69)[a] |
| 38 | 29,376–29,624 | + | 82 | 9.37 | 9.48 | Cro-like repressor | WCS65187.1 | SH-St 15644 (Triavirus) YP_010083143.1 | 100 | 100 | Cro/CI-type_HTH (7–63), HTH_3 (10–58)[a] |
| 39 | 29,637–30,080 | + | 147 | 17.25 | 6.98 | | WCS65188.1 | 3MRA YP_009209302.1 | 100 | 99.32 | – |
| 40 | 30,095–30,274 | + | 59 | 7.13 | 6.72 | | WCS65189.1 | ETA NP_510902.1 | 100 | 100 | – |
| 41 | 30,264–30,503 | - | 79 | 9.43 | 4.76 | | WCS65190.1 | 3MRA YP_009209300.1 | 100 | 100 | – |
| 42 | 30,560–31,351 | + | 263 | 30.28 | 6.26 | Antirepressor protein | WCS65191.1 | 3MRA YP_009209299.1 | 100 | 100 | ANT (149-252), BRO-N (13–107)[c] |
| 43 | 31,352–31,576 | + | 74 | 8.84 | 4.54 | | WCS65192.1 | dv11 (Dubowvirus) NP_803261.1 | 100 | 100 | – |
| 44 | 31,616–32,065 | + | 149 | 17.10 | 7.85 | | WCS65193.1 | dv11 (Dubowvirus) NP_803262.1 | 100 | 100 | – |
| 45 | 32,079–32,294 | + | 71 | 8.17 | 8.03 | Transcriptional regulator | WCS65194.1 | dv11 (Dubowvirus) NP_803263.1 | 100 | 100 | Signal peptide (1–22)[a] |
| 46 | 32,287 32,448 | + | 53 | 6.21 | 5.98 | Putative membrane protein | WVM05245.1 | phiETA (Phietavirus) NP_510908.1 | 100 | 100 | DUF1270 (1–53), TMhelix (7–29)[a], Transmembrane (2)[b] |

**TABLE 1** Bacteriophage ASZ22RN genes and their predicted products (Continued)

| Gene no. | Gene position (bp) | Strand[e] | Length of product (aa) | Predicted molecular mass (kDa) | pI | Known or predicted function | Protein ID | The closest homolog in GenBank (GenBank acc. no.) | Coverage[d] (%) | Identity[d] (%) | Amino acid sequence motifs |
|---|---|---|---|---|---|---|---|---|---|---|---|
| 47 | 32,541–32,843 | + | 100 | 11.17 | 4.93 | | WCS65195.1 | 3MRA YP_009209294.1 | 100 | 100 | DUF2482 (3–99)[a] |
| 48 | 32,848–33,108 | + | 86 | 10.24 | 5.27 | | WCS65196.1 | phiSa2wa_st72 (Triavirus) AUM57962.1 | 100 | 98.84 | DUF1108 (1–84)[a] |
| 49 | 33,118–33,369 | + | 83 | 9.87 | 4.71 | | WCS65197.1 | 3MRA YP_009209292.1 | 100 | 100 | – |
| 50 | 33,362–33,895 | + | 177 | 20.41 | 4.88 | Gam-like protein | WCS65198.1 | 3MRA YP_009209291.1 | 100 | 100 | Phage_Mu_Gam (23–176)[c] |
| 51 | 33,899–34,678 | + | 259 | 29.56 | 6.12 | RecA-like protein Sak4-like ssDNA annealing protein | WCS65199.1 | 3MRA YP_009209290.1 | 100 | 100 | AAA_24 (27–232)[c] |
| 52 | 34,709–35,263 | + | 184 | 21.29 | 5.35 | Single stranded DNA-binding protein Ssb | WCS65200.1 | 3MRA YP_009209289.1 | 100 | 100 | Molibd-lite (137–184)[a] |
| 53 | 35,276–35,968 | + | 230 | 26.78 | 6.23 | Putative HNH-like nuclease | WCS65201.1 | dv11 (Dubowvirus) NP_803267.1 | 100 | 99.13 | HNHc_6 (21–220) |
| 54 | 35,940–36,773 | + | 277 | 32.97 | 6.08 | DNA replication initiation protein | WCS65202.1 | 3MRA YP_009209286.1 | 100 | 100 | EF_HAND_2 (178–213)[a] Phage_rep_org_N (6–125)[c] |
| 55 | 36,786–37,571 | + | 261 | 30.56 | 8.61 | DNA replication helicase loader DnaC/DnaI | WCS65203.1 | 55 YP_240513.1 | 100 | 99.62 | IstB_IS21 (67–257)[a] P-loop NTPase (85–248)[a] |
| 56 | 37,568–37,726 | + | 52 | 6.15 | 5.21 | Putative SapI1 derepressor (1) | WCS65204.1 | 3MRA YP_009209284.1 | 100 | 100 | – |
| 57 | 37,739–37,960 | + | 73 | 8.60 | 5.08 | | WCS65205.1 | phiSLT (Triavirus) NP_075486.1 | 100 | 100 | DUF3269 (1–73)[a] |
| 58 | 37,970–38,374 | + | 134 | 16.08 | 9.43 | RusA-like crossover junction endodeoxyribonuclease | WCS65206.1 | 3MRA NP_075486.1 | 100 | 100 | DUF1064 (1–112)[a] |
| 59 | 38,379–38,564 | + | 61 | 7.29 | 4.10 | | WCS65207.1 | bv77 (Biseptimavi-rus) NP_958649.1 | 100 | 100 | DUF3113 (1–60)[a] |
| 160 | 38,565–39,182 | + | 205 | 23.94 | 9.86 | | WCS65208.1 | 3MRA YP_009209280.1 | 100 | 100 | Coil (151–173)[a] PVL_ORF50 (99–204)[c] |
| 61 | 39,182–39,442 | + | 86 | 9.83 | 5.54 | Putative abortive infection protein | WCS65209.1 | 3MRA YP_009209279.1 | 100 | 100 | SaV-like protein (18–77)[a] DUF3310 (18–78)[c] |
| 62 | 39,445–39,645 | + | 66 | 7.72 | 4.28 | | WCS65210.1 | 3MRA YP_009209278.1 | 100 | 100 | – |
| 63 | 39,660–39,902 | + | 80 | 9.62 | 6.04 | | WCS65211.1 | 3MRA | 100 | 100 | Phage_Orf51 (1–80)[c] |

**TABLE 1** Bacteriophage ASZ22RN genes and their predicted products (Continued)

| Gene no. | Gene position (bp) | Strand[e] | Length of product (aa) | Predicted molecular mass (kDa) | pI | Known or predicted function | Protein ID | The closest homolog in GenBank (GenBank acc. no.) | Coverage[d] (%) | Identity[d] (%) | Amino acid sequence motifs |
|---|---|---|---|---|---|---|---|---|---|---|---|
| 64 | 39,916–40,122 | + | 68 | 8.04 | 5.21 | Putative RBP anchor subunit | WCS65212.1 | bv77 (Biseptimavirus) NP_958654.1 YP_009209277.1 | 100 | 100 | – |
| 65 | 40,125–40,529 | + | 134 | 15.71 | 4.89 | Putative RBP anchor subunit | WCS65213.1 | SMSAP5 (Triavirus) YP_007005703.1 | 100 | 100 | – |
| 66 | 40,526–40,873 | + | 115 | 13.44 | 4.38 | | WCS65214.1 | phi12 (Triavirus) NP_803325.1 | 100 | 100 | YopX protein (4-112)[a] |
| 67 | 40,870–41,259 | + | 129 | 15.45 | 5.20 | | WCS65215.1 | vB_SauS_I73 UKM36431.1 | 100 | 77.52 | Coil (10–65)[a] |
| 68 | 41,252–41,500 | + | 82 | 9.25 | 3.96 | | WCS65216.1 | vB_SauS_760 UKM36226.1 | 100 | 97.56 | DUF1024 (1–82)[a] |
| 69 | 41,493–42,029 | + | 178 | 20.77 | 4.56 | Dimeric dUTPase, SaPIbov1 derepression (1) | WCS65217.1 | phiSa2wa_st1 (Triavirus) YP_010083587.1 | 100 | 100 | dUTPase_2 (9-170)[c] |
| 70 | 42,066–42,272 | + | 68 | 7.92 | 9.52 | | WCS65218.1 | P240 (Triavirus) YP_010083423.1 | 100 | 95.59 | DUF1381 (2–45)[a] |
| 71 | 42,269–42,655 | + | 128 | 14.82 | 9.28 | Putative membrane protein | WCS65219.1 | 3MRA YP_009209271.1 | 100 | 100 | TMhelix (13–36)[a] Transmembrane (1)[b] |
| 72 | 42,652–42,825 | + | 57 | 6.59 | 4.28 | Transcriptional activator RinB (1) | WCS65220.1 | 3MRA YP_009209270.1 | 100 | 100 | RinB (3–53)[c] |
| 73 | 42,826–43,227 | + | 133 | 15.42 | 9.23 | Putative transcriptional activator RinA | WCS65221.1 | ETA NP_510932.1 | 100 | 100 | Coil (53–80)[a] |
| 74 | 43,363–43,455 | + | 30 | 3.37 | 9.11 | Putative membrane protein | WVM05246.1 | vB_SauS_Mh1 OM439673 (17197–17286) | 100 | 100 | TMhelix (7–29)[a] |

[a]Hits from InterProScan (amino acid sequence coordinates).
[b]Number of transmembrane domains from DeepTMHMM 1.0.
[c]Hits from HMMER (amino acid sequence coordinates).
[d]The coverage (in %) and identity (in %) of each protein with its closest homolog from the GenBank database is as calculated with BLASTp.
[e]"+" and "−" indicate the upper and lower DNA strand, respectively.

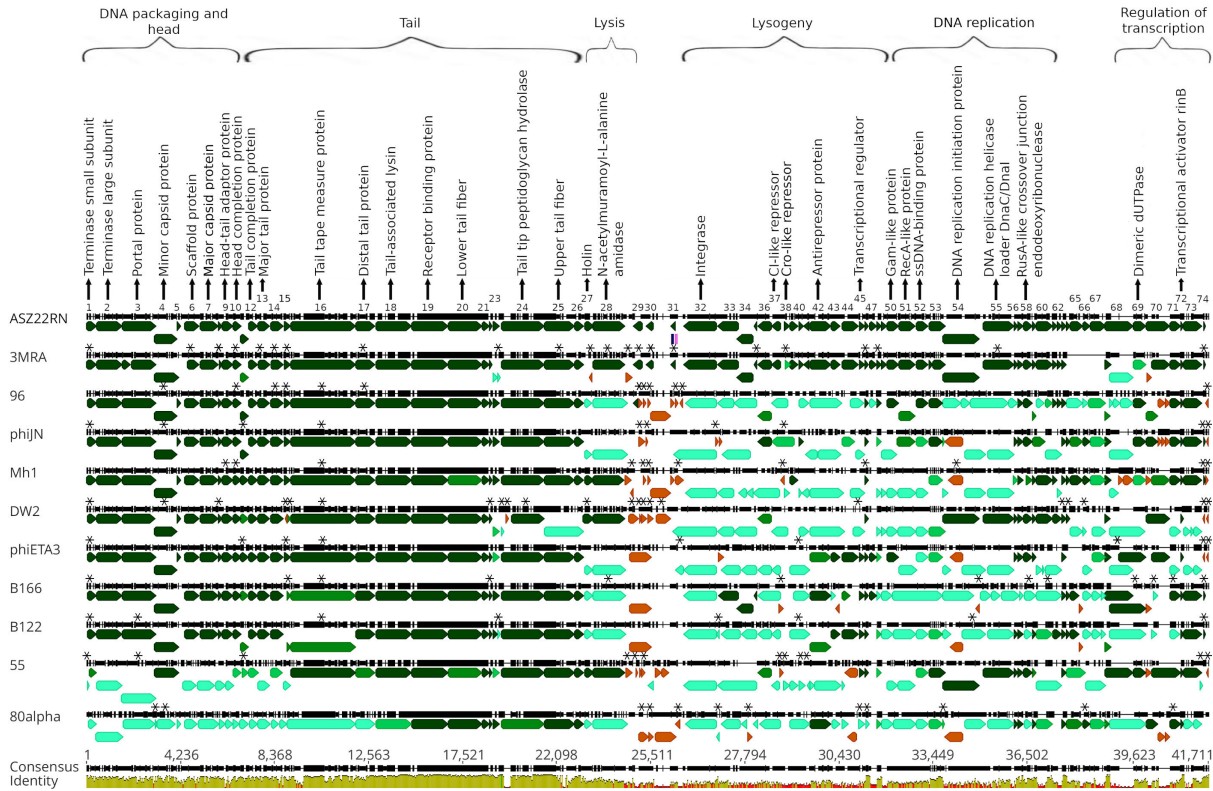

**FIG 3** The alignment of genomic maps of ASZ22RN with the genomic maps of its selected relatives representing *Phietavirus* and *Dubowvirus* genus. Genes in the genomes of ASZ22RN relatives shown in the alignment were indicated based on the GenBank annotations corrected or supplemented according to the analysis performed as described in the Materials and Methods section. The re-annotated or newly identified genes of ASZ22RN relatives are indicated by asterisks. Their coordinates relative to the coordinates of their sequence annotations deposited in GenBank are listed in Table S5. The sequences of some phage genomes in the figure were reorganized for their alignment to set their start positions at the beginning of the small terminase subunit genes. Genes are colored according to the % of the identity of their products to the corresponding products of ASZ22RN genes (dark green: <70%, green: 70 > 40%, blue: 40 > 20%, brown: <20%. The alignment (gapped) was performed with the use of Geneious Prime. The scale length corresponds to the consensus sequence including gaps. The GenBank accession numbers of genomic sequences used for the re-annotations and alignment are provided in Table S5.

dnaC subgroup C2 (dnaC2), dUTPase group 2 (dut2), portal group Bb, serogroup B, and amidase group 1 (ami1) (Table S3). This combination of module types appears to be unique only to phages ASZ22RN and 3MRA from among phages of genomic sequence deposited in GenBank (accessed 20 February 2023). Comparative sequence analysis of ASZ22RN revealed that it is a mosaic of genes of other phietaviruses and phages of related genera with no genes that could be unique to ASZ22RN (Table 1; Fig. 3).

The genome of ASZ22RN encodes 74 proteins (Table 1; Table S4). All of them but four (products of genes 5, 65, 66, and 68) have identical or nearly identical counterparts among predicted products of 3MRA phage proteins, as verified by tblastN (Fig. 3). Similarly to the genomes of other staphylococcal siphoviruses, the ASZ22RN genes can be grouped into a few major functionally conserved modules associated, in order, with DNA packaging and head morphogenesis, tail morphogenesis, cell lysis, lysogeny control, DNA replication, and transcription regulation (Fig. 3). The first genome modules, associated with packaging, head and tail morphogenesis, and cell lysis, encompass 28 genes and are nearly entirely collinear with the relevant modules of the most closely related phietaviruses. Of notion is ASZ22RN gene 4, whose location in the genome concerning other genes is collinear with that of phage 80alpha essential gene 44 of MuF domain-containing protein family. The product of the latter is an ejection (pilot) protein that binds blunt ends of linear DNA in a non-sequence-specific manner and could protect the injected phage DNA from cleavage by exonuclease (29, 30). The

product of ASZ22RN gene 4 is similar in length to 80alpha Gp44, and despite low overall sequence similarities to 80alpha Gp44, it contains the MuF domain motif in its C-terminal moiety (Table 1; Fig. S3). Additionally, as in the case of other phages with MuF domain motif-containing proteins, it is encoded by a gene directly downstream or one gene apart from the portal protein-encoding gene (Fig. 3) (31). Conceivably, its product plays a similar function in the phage DNA ejection as 80alpha Gp44. The arrangement of ASZ22RN genes encoding the tail and baseplate components is typical of staphylococcal siphoviruses with products of genes 16–19 representing in order: the tail tape-measure protein, distal tail protein, tail-associated lysin, and receptor-binding protein.

The cell lysis module of ASZ22RN and other phietaviruses is conserved. It encodes a holin of two transmembrane domains, a feature characteristic for class II holins (32), and a modular endolysin (Table 1; Table S4). The region between the endolysin and integrase-encoding gene of ASZ22RN and related phietaviruses is the most diversified. While in prototypical phietavirus phiETA3 and certain other phietaviruses, it encodes the exfoliative toxin, or homologs of other bacterial toxins, in ASZ22RN and 3MRA, it could encode three small proteins (Gp29, Gp30, and Gp31), which have homologs among predicted proteins of a few other *Phietavirus, Dubowvirus,* or *Triavirus* genus phages, but their function is unknown.

The further part of the ASZ22RN genome is highly similar only to the corresponding part of the 3MRA phage genome, except for certain genome modules. For instance, the modules of DNA replication and transcription regulation of ASZ22RN are also highly similar to those of phage DW2 and 55, respectively, reflecting the horizontal gene transfer between various phietaviruses.

The ASZ22RN 11-gene lysogeny-associated module, which starts from the integrase-encoding gene, encompasses genes encoding transcriptional repressors and the antirepressor. Its first six genes are oriented counterclockwise. Like in certain other phietaviruses, e.g., 3MRA, phiETA, 55, and B166, but unlike in phiETA3, which groups on the same branch of the phylogenetic tree as ASZ22RN (Fig. 2B), the integrase-encoding gene is not followed by any gene encoding a protein with homology to excisionase. Three genes next to it, 33-35, apparently represent a moron—an incremental addition to the phage genome, which is functionally unrelated to the neighboring genes (33). They separate the integrase gene from the other genes of the lysogeny control module. Their predicted products are unique only to ASZ22RN and two other *Phietavirus* genus phages, 3MRA and B166, and to *Triavirus* genus phage 42E (YP_239891.1; 71% coverage, 88% identity). Two of them (Gp33 and Gp35) contain putative transmembrane helices. The predicted structure of Gp33 C-terminal moiety has similarities to the structures of internal moieties of *Serratia marcescens* hemolytic toxin subunits SmhA and SmhB (34) and related pore-forming toxin subunits (Table S4), suggesting similar function. The major part of the ASZ22RN lysogeny control module contains the *immAR* genes, their promoter-operator region, and the divergently transcribed transcriptional regulator gene (35) analogous to phage lambda *cro*. The products of immAR are functionally analogous to the main phage repressor CI, as shown in studies on the prototypical *Bacillus subtilis* ImmA and ImmR of the ICEB*s1* integrative and conjugative element and of phage ø105 (36, 37). Unlike the prototypical CI repressor, which is inactivated by cleavage following the induction of SOS response, the ImmR repressor is inactivated by cleavage by ImmA—a zinc-dependent metalloprotease. The sequence of ASZ22RN ImmA, like ICEB*s1* ImmA, contains the Zn-dependent metalloprotease and HTH motifs, and its predicted structure, like the structure of ICEB*s1* ImmA, is highly similar to the structure of *Deinococcus deserti* metalloprotease and HTH domain of IrrE protein (Table 1; Table S4). The genes encoding ImmR-like and Cro-like repressors are separated from each other by the 163 bp non-coding, common promoter-operator region, which by analogy with other phietaviruses contains the ImmR- and Cro-binding sites and functions as a switch between lysis and lysogeny (15, 16).

The replication module of ASZ22RN is nearly identical to those of phages 3MRA and DW2. It includes 13 genes encoding recombination- and replication-associated

functions. The region between DNA replication helicase loader DnaC/DnaI-encoding gene and the Ruv-like-protein-encoding gene of this module encodes a 52-aa protein (Gp56, WCS65204.1), which shares 49 of 59 amino acid residues with the Sri protein of phage 80alpha. Sri binds to and deactivates SaPI1 Stl repressor (38). The similarity of Gp56 to Sri suggests similar functions. The replication module precedes the module grouping transcription regulation-associated genes. The border region between these two modules contains six genes. In certain related phages, e.g., phiNM4 and DW2, this region encodes virulence-associated proteins (10, 35, 38). Certain ASZ22RN genes of this region are likely to play a similar function. The products of ASZ22RN genes 63 and 65 are nearly identical to the products of phiNM1/phiNM2 and phiNM4 phage genes corresponding to the SAV876 and SAV1978 genes, respectively, of the Mu50 strain prophage. Each of the latter contributes to the virulence of Mu50 for infected nematodes (10, 39).

The transcription regulation module consists of six genes, including the gene encoding the dimeric dUTPase (Gp68) identical to that of phage phiNM4, and to the prophage-encoded virulence factor of *S. aureus* Mu50 strain (SAV0876) (10). The high similarity of the ASZ22RN dUTPase to the dUTPases of phages phiNM1 and phiNM2, which can derepress SaPIbov1 pathogenicity island (38), suggests that ASZ22RN can also function as the SaPIbov1 derepressor. The two genes immediately downstream of the dUTPase gene have counterparts in the genomes of certain other staphylococcal siphoviruses, but their functions are unknown. They precede two genes that, by prediction, encode transcription-activating proteins RinB and RinA. While ASZ22RN RinB is highly similar to RinB proteins of certain other *S. aureus* siphoviruses, the ASZ22RN RinA has no significant sequence similarities to known RinA proteins of proven function. However, like in the case of characterized RinA proteins, the predicted structure of its C-terminal moiety is significantly similar to the structure of conserved region 4 of several RNA polymerase sigma factors, responsible for specific binding to promoter elements, which justifies its functional assignment (Table S4; 40). Despite the huge diversity of RinA protein sequences between various staphylococcal siphoviruses, the function of these proteins as transcriptional activators of phage packaging, assembly, and lysis genes is conserved (40). The last gene of the transcription regulation module encodes a small, conserved putative membrane protein of unknown function. None of the ASZ22RN-encoded proteins is similar to SaPIbov2 derepressor (ABF71586) encoded by phage 80alpha (38).

## ASZ22RN prophage integration site

Comparison of genomic sequences of enrichment culture strains with the sequence of ASZ22RN revealed that the source of ASZ22RN phage was a prophage of 4472/08 strain, which represents the methicillin-resistant *S. aureus* (MRSA) strain of clonal complex 7 (CC7) isolated from human blood. DNA sequences of the ASZ22RN phage and a prophage integrated with the genome of this strain are identical. The borders of these two sequence identities in the genome of the 4472/08 strain indicated the bacterial attachment site (*attB*) for ASZ22RN prophage and the phage attachment site (*attP*) in the ASZ22RN genome (Fig. S4). The *attP* and *attB* of ASZ22RN appeared to be identical to those of staphylococcal *Dubowvirus* genus phages φ11 [NC_004615.1] and phiNM1, and to staphylococcal *Peeveelvirus* phage phiPV83-pro (Fig. S4) (10, 11, 41). The regions of two tandem direct repeats involved in the recognition by integrase are identical in ASZ22RN and φ11 [NC_004615.1]. Regions identical to ASZ22RN *attP* and the integrase binding site can be also found in 14 other staphylococcal siphoviruses representing three different genera: *Phietavirus* (3MRA, Sushi, vB_SauS_I73, 88, 29, 187, and vB_SauS_SAP27 [GenBank acc. no.: NC_028917.1, ON571632.1, OM439672.1, NC_007063.1, NC_007061.1, NC_007047.1, and NC_054981.1, respectively]), *Dubowvirus* (phiMR25, 69, and MSP1 [GenBank acc. no.: NC_010808.1, NC_007048.1, and OQ851348, respectively]) and *Peeveelvirus* (B_UFSM1, B_UFSM3, B_UFSM4, and B_UFSM5 [GenBank acc. no. MW650841.1, MW627293.1, MW147366.1, and MW192778.1, respectively]).

The ASZ22RN prophage integration disrupted the *S. aureus yfkA* gene between nucleotides 30 and 31 of the YfkA protein coding sequence. However, we noted that the prophage terminal region proximal to the remaining part of the *yfkA* gene could encode, by prediction, an alternative 16-aa, N-terminal region of YfkA that could potentially replace the original 29-aa N-terminus of this protein (Fig. S5A). Eight amino acid residues of this region are identical to those at the relevant region of *S. aureus* YfkA protein, and one represents a conservative replacement. Four nucleotide residues upstream of the alternative beginning of the *yfkA* gene, there is a sequence (TAAGGAGTTTATA) that resembles the typical staphylococcal Shine-Dalgarno region in mRNA and complementary to the 3′ end of ribosomal 16S rRNA (underlined bases) which is involved in the initiation of translation. This region is preceded by sequences resembling −35 and −10 elements of promoters for the housekeeping RNA polymerase (Fig. S5B). Further studies will be required to confirm whether cells harboring the ASZ22RN prophage, or prophages with related *attP* and the integrase binding sites, produce the alternative form of YfkA protein.

## Host range of ASZ22RN

Testing the sensitivity of 47 *S. aureus* strains representing 12 clonal complexes (CCs) to infection with ASZ22RN revealed that the phage can productively infect only nine of these strains, as indicated by the appearance of plaques (Table 2). Productive infection concerned certain strains of clonal complexes 1, 5, 8, 22, and 45, and in no case all strains of a given clonal complex. It was most efficient in the case of CC5 and CC8 strains. In the case of other *S. aureus* strains tested, one could see a lysis zone under a drop of highly concentrated phage suspension but no single plaques with diluted phage suspensions. The phage used for strain sensitivity testing was purified from cell-derived lysins or bacteriocins by washing. This indicates that the observed lysis zones resulted from phage-mediated lysis from without, a phenomenon caused by phage adsorption to bacteria at a threshold limit and associated with the contents liberation by a distension and destruction of the cell wall (42, 43). Clearly, cell envelopes of all tested strains that cannot support productive infection of ASZ22RN can bind ASZ22RN and can be penetrated by ASZ22RN cell puncturing machinery, owing to the action of phage tail tip complex which was described in detail for phage 80alpha, closely related to ASZ22RN (44) (see also Fig. 3). The concentration of phage needed to cause lysis from without was markedly different in the case of different strains. In most cases, phage suspensions of $10^6$–$10^{10}$ PFU/mL caused lysis from without. In the case of CC398 strain 463/10, a semitransparent lysis zone could be seen on a layer of 463/10 cells only when phage suspension of at least $10^{12}$ PFU/mL was used for testing. Two strains of other *Staphylococcus* species tested, *S. hyicus* DSM-17421 and *S. lugdunensis* DSM-4804, were resistant to infection with ASZ22RN (Table 2).

## Potential of ASZ22RN in plasmid transfer by transduction

The ability of ASZ22RN to productively infect or lyse from without all tested *S. aureus* strains, independent of the clonal complex represented by a given strain, prompted us to test whether ASZ22RN can be suitable for the efficient transfer of a shuttle *Escherichia coli-S. aureus* plasmid, pMLE5, to these strains. Transduction of the plasmid from a pMLE5-containing derivative of the RN4220 strain to a plasmid-free RN4220 could occur but with low efficiency (Fig. 4). Typically, when 0.5 mL of dense cell suspension was mixed with 0.5 mL of phage suspension ($10^9$ PFU/mL), only a few colonies of plasmid-carrying transductants could be recovered upon plating 1/10th of the mixture on a selective plate. However, the efficiency of transduction was increased by up to nearly five orders of magnitude if the plasmid was supplemented with the 495 bp fragment of ASZ22RN DNA encoding the small terminase subunit (TerS) and containing a preferred *pac* region of ASZ22RN. Surprisingly, the modified plasmid (pLKA18) could be transduced from RN4220 to all tested randomly selected clinical isolates of *S. aureus*. The efficiency of transduction to strains that were productively infected with ASZ22RN was the highest,

**TABLE 2** Sensitivity of various *Staphylococcus* strains to bacteriophage ASZ22RN[a]

| Clonal complex | Strain | Lysate dilutions | | | | | |
|---|---|---|---|---|---|---|---|
| | | $10^0$ | $10^{-1}$ | $10^{-2}$ | $10^{-3}$ | $10^{-4}$ | $10^{-5}$ |
| CC1 | 1203/05 | CL | TL | – | – | – | – |
| | 1693/05 | CL | TL | MP | – | – | – |
| | 3572/09 | CL | TL | – | – | – | – |
| | 717/05 | CL | CL | TL | – | – | – |
| CC5 | 300/07 | CL | CL | CL | CL | TL | UP-M |
| | 1034/05 | CL | TL | TL | MP | UP | UP |
| | 1793/05 | CL | TL | TL | MP | UP | UP |
| | 1880/05 | CL | TL | TL | – | – | – |
| | 2875/05 | TL | TL | – | – | – | – |
| | 4000/07 | CL | TL | TL | – | – | – |
| | 10250/11 | CL | TL | TL | – | – | – |
| CC7 | 1171/05 | TL | TL | – | – | – | – |
| | 1268/05 | CL | TL | – | – | – | – |
| | 4472/08 | TL | TL | – | – | – | – |
| | 10525/11 | TL | TL | – | – | – | – |
| CC8 | 2065/05 | CL | CL | CL | TL | MP | UP |
| | 2261/05 | CL | CL | CL | TL | TL | UP-M |
| | 2945/06 | TL | TL | – | – | – | – |
| | 3009/17 | CL | CL | TL | TL | – | – |
| | 10174/99 | TL | TL | – | – | – | – |
| | 10415/11 | CL | CL | CL | TL | TL | UP-M |
| CC8/239 | 4065/07 | CL | TL | TL | TL | – | – |
| | 2262/05 | CL | CL | TL | TL | – | – |
| | 4124/07 | TL | – | – | – | – | – |
| | 101/00 | CL | CL | TL | – | – | – |
| CC9 | 2064/05 | TL | TL | – | – | – | – |
| | 3864/05 | TL | TL | – | – | – | – |
| CC15 | 1700/07 | TL | – | – | – | – | – |
| | 1881/05 | TL | – | – | – | – | – |
| | 2929/07 | TL | – | – | – | – | – |
| CC22 | 706/05 | CL | CL | UP-M | UP | 2 | – |
| | 6068/10 | CL | TL | TL | – | – | – |
| CC30 | 2584/01 | CL | TL | TL | – | – | – |
| | 1528/09 | TL | – | – | – | – | – |
| | 4341/10 | TL | – | – | – | – | – |
| CC45 | 247/07 | CL | TL | TL | – | – | – |
| | 1286/05 | TL | TL | – | – | – | – |
| | 1105/06 | CL | CL | TL | UP | 15 | – |
| | 1452/05 | CL | TL | – | – | – | – |
| | 1786/05 | CL | TL | – | – | – | – |
| | 2795/06 | CL | TL | – | – | – | – |
| CC59 | 140/05 | CL | TL | TL | – | – | – |
| | 1781/05 | CL | TL | – | – | – | – |
| CC398 | 5069/08 | TL | TL | – | – | – | – |
| | 7202/12 | TL | TL | – | – | – | – |
| | 463/10 | – | – | – | – | – | – |
| | 5074/08 | TL | – | – | – | – | – |
| *S. hyicus* | DSM-4804 | – | – | – | – | – | – |
| *S. lugdunensis* | DSM-17421 | – | – | – | – | – | – |

[a]The strains that are productively infected with ASZ22RN are marked with gray. Phage for the assays ($6 \times 10^{10}$ PFU/mL) was purified from the bacterial lysins and bacteriocins by washing, as described in the Materials and Methods section. CL, clear lysis; TL, turbid lysis; MP, merged plaques; UP, uncountable plaques; UP-M, uncountable-merged plaques; –, no lysis; numbers indicate the number of plaques.

and nearly as high as that to the RN4220 strain. The efficiency of transduction to strains that could not be productively infected but lysed from without was two to four orders of magnitude lower but still high enough to obtain hundreds of transductants upon plating the diluted mixtures of recipient bacteria with the phage. Surprisingly, the plasmid could be transferred by ASZ22RN even to the 463/10 strain. Moreover, the efficiency of its transduction to this strain was comparable to that of strains that could be lysed from without when treated with ASZ22RN suspensions of phage titers that were a few orders of magnitude lower than that required to see the turbid lysis zone on a layer of 463/10 cells.

## Analysis of DNA content in ASZ22RN-transducing particles

To analyze the DNA content of ASZ22RN-transducing particles, the phages isolated from the cells of RN4220 strain with pMLE5 or pLKA18 plasmid were sequenced using Illumina and MinIon nanopore technology, and the obtained sequence reads were compared with the sequences of pMLE5 or pLKA18 plasmid and RN4220 strain. Of a total of 38,658 MinIon reads of over 1 kb in length representing DNA isolated from the RN4220/pMLE5 strain, only two, 2,844 bp and 1,688 bp in length, matched the sequence of the pMLE5 plasmid (Table S6). Of a total of 136,002 reads representing DNA isolated from the RN4220/pLKA18 strain, 477 reads matched the pLKA18 sequence, indicating that pLKA18 DNA is packed to the ASZ22RN virions at least two orders of magnitude more frequently than pMLE5 DNA.

MinIon reads that matched the pMLE5 sequence and were of the length of ASZ22RN DNA fitting to the capacity of the ASZ22RN capsid (42–46 kb) were not detected among the DNA molecules of ASZ22RN phage propagated in RN4220/pMLE5 cells. MinIon reads that matched the pLK18 sequence were represented by 16 of a total of 12,600 analyzed reads of ASZ22RN DNA length among the DNA molecules of ASZ22RN phage

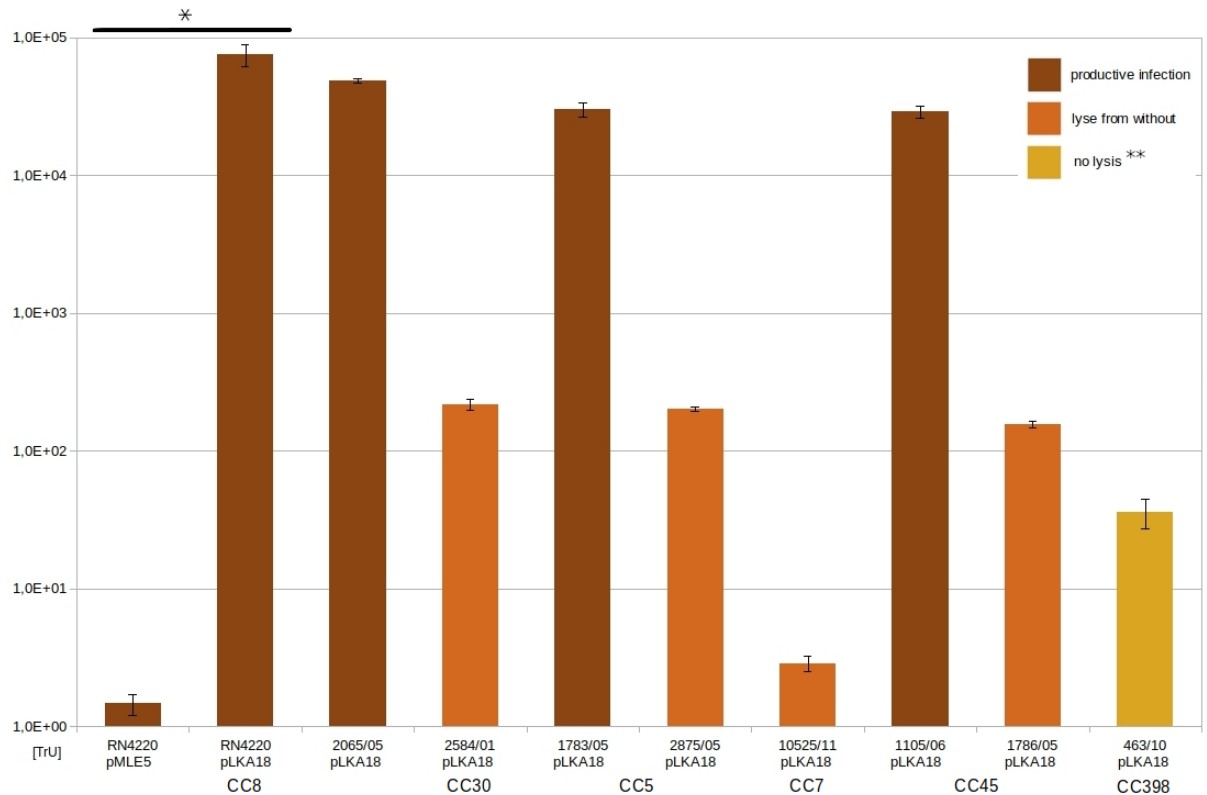

**FIG 4** Efficiency of transduction (TrU) of pMLE5 plasmid and pLKA18 by phage ASZ22RN. *Student's *t*-test *P*-value = 0.04. **Lysis from without was observed only with the concentrated phage suspension ($10^{12}$/PFU/mL).

propagated in RN4220/pLKA18 cells (Fig. 5; Table S6). Ten of the pLKA18-containing DNA molecules analyzed were concatamers composed solely of pLKA18 genomic units, and five of them started from various sites within the ASZ22RN *terS* gene. The remaining six sequences were hybrids between pLKA18 and ASZ22RN DNA and contained various numbers of pLKA18 genomic units. In all of them, the junction between pLKA18 sequence and ASZ22RN sequence was within the *terS* gene, indicating *terS* as the recombination region between pLKA18 and ASZ22RN DNA. To verify whether the intact plasmid was reconstructed in the chloramphenicol-resistant transductants, we compared, by agarose gel electrophoresis, the migration pattern of DNA molecules isolated from a transductant with the migration pattern of DNA molecules isolated from a transformant of RN4220 cells with the pLKA18 plasmid (Fig. S6). They were similar.

The fraction of MinIon and Illumina reads representing RN4220 DNA was similar in the preparations from cells containing the pMLE5 and pLKA18 plasmid (compare results in Tables S6 and S7). Thus, they were grouped to analyze their content. As many as 0.11% of all Illumina reads and 0.14% of MinIon reads represented RN4220 DNA. They were preferentially mapped to certain regions of the RN4220 genomic sequence (Fig. S7). The region of the highest coverage adjoined the left border of the ASZ22RN attachment from which the coverage was gradually decreasing up to about 1 MB. Other parts of the genome were also represented in the transducing particles, albeit with much lower frequencies than the region to the left of the ASZ22RN attachment site. This pattern of RN4220 genomic DNA distribution in transducing particles is consistent with the ability of certain staphylococcal siphoviruses to transfer DNA via lateral and generalized transduction that was previously described as a feature of *Dubowvirus* genus phages, including phage ϕ11 (25).

## A search for the immunity determinants against ASZ22RN in strains that are lysed from without but are not productively infected with phage ASZ22RN

The ability of *S. aureus* strains that could not support ASZ22RN infection but were lysed from without, to serve as recipients in the ASZ22RN-mediated transduction of plasmid DNA, indicated that DNA can be introduced to the cells of these strains by ASZ22RN infection, but further lytic development of the phage is prevented. This prompted us to search for the potential immunity determinants against ASZ22RN in the genomes of these strains. Our search with the use of DefenseFinder (see

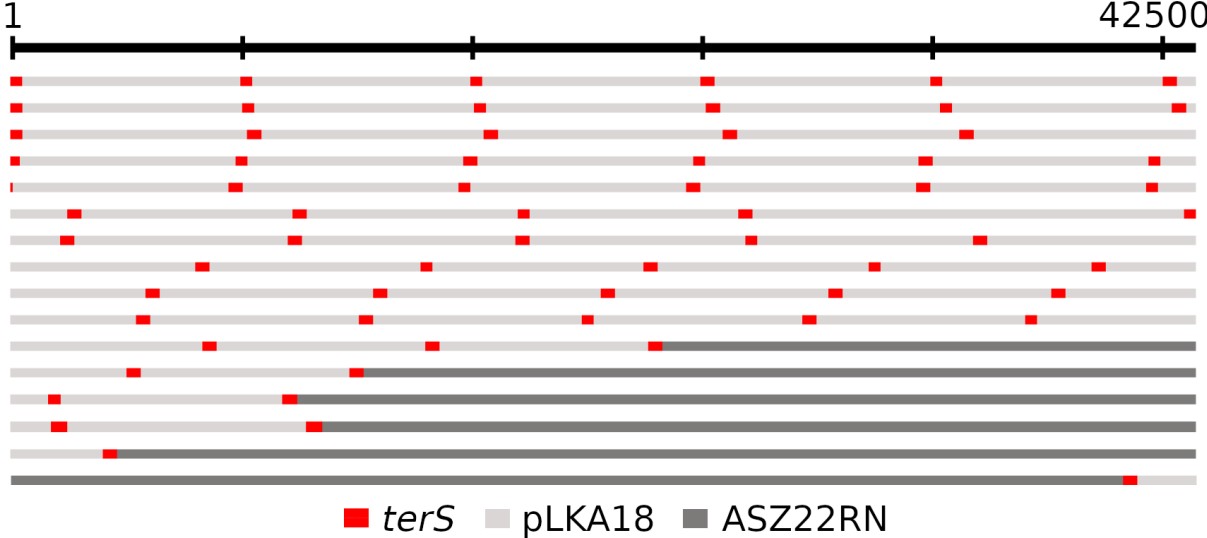

**FIG 5** Composition of DNA molecules in pLKA18 plasmid-transducing particles of bacteriophage ASZ22RN. DNA of ASZ22RN for sequencing was isolated from phages propagated in RN4220/pLKA18. Fragments representing DNA of the pLKA18 plasmid and ASZ22RN phage are shown in light or dark gray, respectively, except for the *terS* gene, which is shown in red.

```
ASZ22RN     29213   TTCAAATTTACCTCCGTTTTATTTATAACAGTATAATAACACTTTTCCATATAGGAAACA    29273
4472/08   2058219   -----------------------------------------------------------    2058159
2875/05   1986536   -----------------------------------------------------------    1986477
4124/07   1956644   -----------------------------------------------------------    1956585
463/10     396616   -----------------------------------------------------------     396675
1880/05   1606175   -----------------------------------------------------------    1606116
5074/08    366265   -----------------------------------------------------------     366324
7202/12    320239   -----------------------------------------------------------     320298
5069/08   1595535   -------------------------T---------------------------------    1595476

ASZ22RN     29274   ACTAGCATTTTAAAAGAATAAAAAATATTTTTCGAGATTTTTGTTGACAATTAGGAAACT    29333
4472/08   2058159   -----------------------------------------------------------    2058100
2875/05   1986476   -----------------------------------------------------------    1986417
4124/07   1956584   -----------------------------------------------------------    1956525
463/10     396676   -----------------------------------------------------------     396735
1880/05   1606115   -----------------------------------------------------------    1606056
5074/08    366325   -----------------------------------------------------------     366384
7202/12    320299   -----------------------------------------------------------     320358
5069/08   1595475   ----------------G------------------------------------------    1595416

ASZ22RN     29334   TAGGTTTAGTATTGAGTTAACTTCAAAAAACGGAGGTGAGCAA    29375
4472/08   2058099   -------------------------------------------    2058057
2875/05   1986416   -------------------------------------------    1986374
4124/07   1956524   -------------------------------------------    1956482
463/10     396736   --A-----A----------------------------------     396778
1880/08   1606055   --A-----A----------------------------------    1606013
5074/08    366385   --A-----A----------------------------------     366427
7202/12    320359   --A-----A----------------------------------     320401
5069/08   1595415   -GA-----A----------------------------------    1595373
```

**FIG 6** Alignment of the 163 bp DNA sequence of phage ASZ22RN region between genes *immR* and *cro* with the relevant prophage sequences identified in certain tested *S. aureus* strains resistant to productive infection with ASZ22RN.

Materials and Methods section) did not reveal any clear correlation between the presence/absence of a phage resistance determinant against a given group of phages in the genomes of particular strains and the ability/inability of phage ASZ22RN to productively infect these strains. Four strains (300/07, 1034/05, 1793/05, and 1528/09) appeared to encode serine/threonine protein kinases Stk2 (WP_0010001347, genome coordinates 84956–86462, 84078–85583, 84214–85539, and 84034–85539, respectively) and Stk1 (WP_000579570.1; genome coordinates 1292741–1294732, 1200322–1202313, 1244400–1246391, and 1244400–1246391, respectively). In previous studies, Stk2 was shown to provide immunity against all five tested staphylococcal siphoviruses, related to ASZ22RN (*Phietavirus* phiNM4, and dubowviruses 80alpha, 85, phiNM1 and phiNM2) in the abortive infection pathway involving Stk1 (45). However, only one of our strains encoding Stk1 and Stk2 (1528/09) could not be productively infected with ASZ22RN.

Eleven strains, among them 10 that could not be productively infected with ASZ22RN (101/00, 2262/05, 2945/06, 4065/07, 10174/99, 463/10, 5069/08, 5074/08, 7202/12) and one that could be infected productively (2261/05) contained genomic regions with similarities to CRISPR-Cas III phage defense system-encoding loci, but their predicted spacer sequences had no significant similarities to any fragments of the ASZ22RN genomic sequence.

Surprisingly, 8 out of 38 *S. aureus* strains lysed by ASZ22RN from without but not infected productively, excluding the ASZ22RN lysogen (strain 4472/08), contain in their DNA the regions identical (three strains) or nearly identical (two to five mismatches, five strains) to the lysis-lysogeny control region between the ImmR- and Cro-encoding genes of ASZ22RN (Fig. 6). Additionally, the predicted products of ImmR and Cro genes of prophage sequences flanking these regions are identical or nearly identical to the ImmR and Cro proteins of phage ASZ22RN. Lysis-lysogeny control regions of staphylococcal

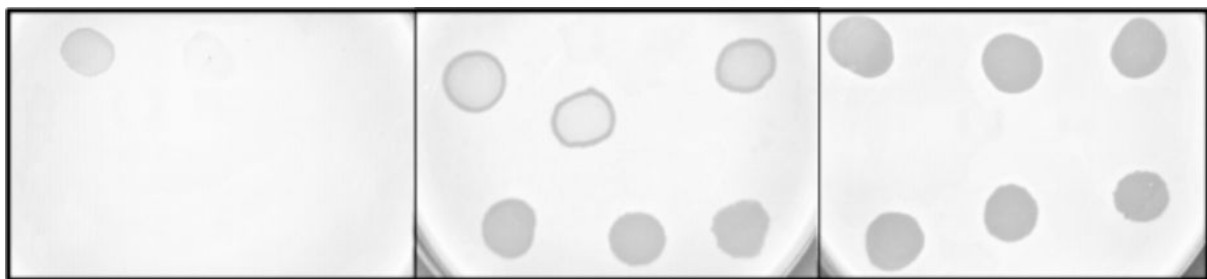

**FIG 7** Influence of cloned *immR* or *immRA* genes of phage ASZ22RN on the susceptibility of *S. aureus* RN4220 cells to ASZ22RN infection. Serially diluted lysates of phage ASZ22RN (initial titer: $10^{10}$ PFU/mL) were used in spot tests on lawns of *S. aureus* RN4220 cells harboring either the *immR* gene (left panel) or the *immRA* gene cluster (central panel), both including their upstream intergenic region cloned into the pMLE5 plasmid. Wild-type RN4220 cells served as a control (right panel). Plates were incubated overnight at 37°C prior to result assessment. To enhance lysis zone visibility, image colors were reversed and desaturated.

siphoviruses, like that of canonical lambda phage, in addition to their role in switching between the lytic and lysogenic mode of phage propagation, are also immunity determinants (15, 16, 46–49). This implies that 21% of *S. aureus* strains that could not support productive infection of ASZ22RN in our study are immune to this phage due to the binding of their prophage repressors present in the cytoplasm with the region between the *immR* and *cro* genes of injected ASZ22RN DNA and blocking ASZ22RN lytic development. Consistently, when we introduced the cloned ASZ22RN *immR* gene to cells of the RN4220 strain, the cells of transformants acquired immunity to infection with phage ASZ22RN (Fig. 7). When the cloned DNA fragment contained, in addition to *immR*, the *immA* gene that by prediction encodes the ImmR-inactivating protease, the resistance of transformants to the infection was not observed.

## DISCUSSION

Increasing problems with curing infections with antibiotic-resistant *Staphylococcus aureus* strains are a driving force in studies on this bacterium and in the development of molecular biology tools for functional genomics of *S. aureus* and the identification of new antibacterial targets. Sources of such tools and also contributors to the *S. aureus* pathogenicity are plasmids and bacteriophages. Many of the latter have the ability to transfer genetic material between *S. aureus* strains by transduction (2, 50). Here, we characterize a new temperate *S. aureus* phage ASZ22RN of features typical to staphylococcal siphoviruses. We demonstrate its transducing potential in transfer plasmids between *S. aureus* cells, provide the analysis of DNA in plasmid and host DNA-transducing particles, and show limitations for ASZ22RN productive infection that do not limit the transducing potential of ASZ22RN.

Comparative analysis of the ASZ22RN genome revealed that it represents a new species within the *Phietavirus* genus. However, among the 51 phages currently classified to phietaviruses (according to GenBank accessed on 3 June 2024), only one phage, 3MRA (51), appears to be highly similar to ASZ22RN, suggesting that ASZ22RN and 3MRA may form a separate clade of phietaviruses.

A common feature of ASZ22RN and 3MRA is their origin from *S. aureus* MRSA strains isolated from the blood of human subjects with invasive bloodstream infection (Table S1) (51). Both phages share a few proteins with significant similarities to virulence-associated factors such as pore-forming toxin SmhA or hemolysin subunit HblL1 (Gp33; Table S4; 34, 52), as well as a virulence-associated protein of phages phiNM1/phiNM2, related to ASZ22RN (Gp63) (Fig. 3) (10). This suggests similarities between pathogenicity-associated traits of strains lysogenized with ASZ22RN or 3MRA. However, not all virulence factor-like proteins encoded by ASZ22RN have homologs encoded in 3MRA. Examples are Gp65 similar to virulence factor of phiNM4 phage (10) and Gp66 similar to YopX domain protein—a *Yersinia* effector delivered to the cytoplasm of a host and encoded also by

staphylococcal and enterococcal phages (53–55). Presumably, the impact of ASZ22RN and 3MRA prophages on their host pathogenicity is not exactly the same.

Isolation of ASZ22RN from a mixed culture of *S. aureus* strains, including the ASZ22RN lysogen, indicates that the ASZ22RN prophage can be spontaneously induced, like 3MRA (51). Both ASZ22RN and 3MRA prophages were also induced by mitomycin C, a known trigger of SOS response. The induction has to occur via the products of the ASZ22RN and 3MRA *immAR* regulatory modules, which are identical. We prove here that ASZ22RN *immR* encodes the repressor of phage lytic development and that the product of *immA* counteracts this repression (see Fig. 7). Regulatory modules similar in sequence to ASZ22RN *immRA* are present in the genomes of nearly 24% of sequenced staphylo-coccal phages and in the SapI3 pathogenicity island (56). The counterpart of ASZ22RN and 3MRA ImmR encoded by SapI3 is a substrate for proteolytic cleavage by SapI3 ImmA. In *B. subtilis* ICEBs1, the ImmR-activating ImmA protease is activated by RecA or a two-protein signaling system (36, 37). The inducibility of ASZ22RN and 3MRA prophages by mitomycin C also suggests the RecA-mediated activation of ImmA. The induction of 3MRA was inhibited by human blood, which was associated with higher expression of the phage repressor gene. Whether the ASZ22RN *immRA* module undergoes similar regulation, and what activates ASZ22RN ImmA, remains to be found.

A search of complete *S. aureus* genomic sequences deposited in GenBank (accessed 3 June 2024) revealed that only one of them (UNC_SaCF36, CP089154), representing a strain from a murine chronic infection model, contains a prophage nearly identical in sequence (98% coverage, 99.98% identity) to ASZ22RN. This contrasts with the common occurrence of lysis-lysogeny switch regions identical or nearly identical in their repressor binding sequences and in the sequences of encoded ImmR-like and Cro-like repressors to those of ASZ22RN in 14 phages or prophages of *Phietavirus*, *Dubowvirus*, *Triavirus*, or *Peevelvirus* genera, and in the prophage regions of 167 *S. aureus* genomes deposited in GenBank (as accessed in 22 October 2024) (Tables S8, S9 and S10). One can predict that the lysogens of the aforementioned phages and the aforementioned strains are immune to infection with ASZ22RN and other phages of identical lysis-lysogeny switch module. Consistent with this prediction, we found here that 8 out of 38 *S. aureus* strains that are not ASZ22RN lysogens and cannot support productive infection with ASZ22RN contain a prophage with the lysis-lysogeny control region identical to that of ASZ22RN. None of the tested strains that were productively infected with ASZ22RN contained this region. Taken together, our data indicate that genome modules containing specific lytic phage repressor genes and their binding sites contribute to the resistance to particular siphoviruses in a significant fraction of *S. aureus* strains and have been exchanged by horizontal gene transfer not only between phages representing different genera of the *Azeredovirinae* subfamily, but also between phages of the *Azeredovirinae* subfamily and the *Bronfenbrennervirinae* subfamily. Taking into account that several *S. aureus* strains are polylysogens, one can predict that one of the dominant mechanisms of resistance to siphoviruses in *S. aureus* is superinfection exclusion.

We show here that ASZ22RN integrates in the 5′ terminus of the *yfkA* gene encoding a radical SAM/CxCxxxC motif-containing protein of unknown function (HDC9079853), and that the left end of the integrated prophage could encode, by prediction, an alternative N-terminus of YfkA protein similar in sequence to that of YfkA, suggesting the reconstitu-tion of functional *yfkA* gene in lysogens. Several staphylococcal prophages that insert into protein-coding genes cause negative lysogenic conversion, which is manifested by the change in the phenotype of lysogens (2). Examples of functional reconstitution of protein-coding genes interrupted by prophage insertion, by providing an alternative fragment of the interrupted gene from a prophage, are known only in the case of certain phages lysogenizing bacteria of other genera, like e.g., *Brochothrix thermosphacta* and *Listeria monocytogenes* (57, 58).

Phage ASZ22RN was able to transfer plasmid and chromosomal DNA by transduction. In this respect, it is similar to certain other staphylococcal phages that pack their DNA by a headful mechanism (3; summarized by 6, 59). While the transduction frequency

of the low-copy-number shuttle *E. coli-S. aureus* 8 kb plasmid vector used in our study was low, it increased a few orders of magnitude when the vector was enriched with the *terS* gene of phage ASZ22RN (Fig. 4). The increase in transduction frequency of plasmids containing a cloned fragment of transducing phage DNA was observed previously in the case of staphylococcal phage φ11 (60, 61). In that case, the analysis of transducing particles by DNA-DNA hybridization revealed that they contained mostly tandem repeats of the plasmid-phage chimera and required plasmid-initiated replication to be formed (60). Our plasmid vector replicates in *S. aureus* via the theta mode, and the copy number of its *S. aureus* plasmid core was estimated to be about 5 (62, 63). The high transduction efficiency of its *terS*+ derivative and the predominance of solely plasmid concatamers in most of the plasmid-transducing particles of ASZ22RN (see Fig. 5) demonstrate that transduction offers a more efficient alternative to transformation for plasmid delivery to *S. aureus* cells, without the risk of co-infection with phage DNA.

All *S. aureus* strains tested in this work could serve as recipients for ASZ22RN-mediated plasmid transduction, independent of their ability to support ASZ22RN productive infection. Thirty-nine of the tested strains, including the ASZ22RN lysogen, could be only lysed from without, indicating that the progression of infection is blocked after the penetration of cell envelopes by the phage. The efficiency of transduction was higher in strains that could be productively infected with ASZ22RN compared to those that were lysed from without. However, the efficiency of lysis from without did not fully correlate with the efficiency of transduction. For instance, cells of the 463/10 strain were transduced with about one order of magnitude higher efficiency than cells of the 10525/11 strain, despite the fact that cells of the latter were lysed from without by the phage at a concentration about five orders of magnitude lower than that required for the lysis of the former (Table 2; Fig. 4). The transducibility of the 463/10 strain may be related to a phenomenon originally described as auto-transduction, in which a prophage-containing strain acquires genetic material captured by transducing particles of a phage that was spontaneously induced from its own prophage and occasionally packed heterologous DNA by propagation in another susceptible strain (64). The prophage repressor of the 463/10 strain protects this strain from productive infection with phage ASZ22RN by binding to the lysis-lysogeny control region in the injected ASZ22RN DNA, which is similar to that of the prophage in the 463/10 strain. However, it apparently cannot prevent the entry and establishment of plasmid DNA transduced by the virions of ASZ22RN.

The ability of staphylococcal-transducing phages to transfer certain small- and medium-sized staphylococcal plasmids to strains that could not be productively infected with these phages was demonstrated previously, but the recovery of transductants was very low (65). In our experiments, the shuttle, medium-size plasmid, was transduced to recipient cells insensitive to productive phage infection with so high efficiency that hundreds of transductants could be easily obtained. Additionally, our results show that a simple test—the sensitivity of a given strain to lysis from without by a given phage—could be used as a predictor of this strain's ability to serve as a recipient in transduction.

The penetration of cell envelopes by staphylococcal siphoviruses depends on virion structures responsible for phage adsorption, digestion of peptidoglycan, and formation or the usage of pores in the cytoplasmic membrane for the passage of virion DNA (66). The 632-aa receptor binding protein of phage ASZ22RN (Gp19, WCS65171.1) is 91% identical to that of staphylococcal phage φ11 (Gp45; NP_803298.1, 636 aa), which has been analyzed in detail (67, 68). The 490-aa central and C-terminal part of ASZ22RN Gp19 shares 486 amino acid residues with the corresponding part of φ11 Gp45, which is involved in the recognition and binding to a host cell receptor. Additional likely ASZ22RN virion components required for phage adsorption and/or the passage of virion-encapsidated DNA are products of genes 24 and 25. They correspond to φ11 products of genes 49 (tail-associated cell wall hydrolase) and 50 (upper tail fiber), respectively. Genes 49 and 50 were the only ones of 28 analyzed genes of φ11, whose knockouts severely impacted

plasmid transfer without a significant influence on phage titer and the transfer of SaPI (69).

Gp45 of phage φ11 binds to the *S. aureus* cell wall with α- or β-O-GlcNAc modified wall teichoic acid(WTA) regardless of the O-GlcNAc anomeric configurations (68). Lipoteichoic acid cannot serve as a receptor (70). High similarity between φ11 Gp45 and ASZ22RN Gp19 suggests a similar requirement for the adsorption of these phages.

The injected DNA is vulnerable to foreign DNA-directed cell defense mechanisms. It is generally accepted that horizontal gene transfer between *S. aureus* strains is limited mainly by types I and IV restriction-modification (R-M) systems, which differ in specificity in strains of different CCs (71–73). They practically block plasmid transfer between *S. aureus* strains of different CCs by transformation. The ability of ASZ22RN phage propagated in the laboratory R-M RN4220 strain to introduce foreign DNA to strains of different CCs implies that introduced DNA is protected from restriction enzymes of different specificity. The ASZ22RN DNA contains multiple sites recognized by various *S. aureus* R-M enzymes of type I (Table S11). Hence, the phage had to evolve other mechanisms of protection from restriction than the paucity of recognition sites for restriction endonucleases. Additionally, the presence of multiple sites for type I restriction enzymes in the pLKA18 plasmid, which is transduced to cells of different CCs by ASZ22RN, suggests that protection is provided to any virion-encapsidated DNA, most likely by a virion protein(s), as in the case of enterobacterial phage P1 (74, 75). A candidate antirestriction protein of ASZ22RN is the product of gene 4. We have argued in this work that it is a likely counterpart of the phage 80alpha product of essential gene 44a protein ejected to cells with virion DNA at infection and protecting phage DNA from nucleases in a sequence-independent manner (Table 1; Fig. S3) (29, 30). Additionally, even if plasmid concatamers injected into cells by transducing phage particles are not protected from host restriction, their remnants may still serve as substrates for homologous recombination, enabling a recovery of functional plasmids.

Unexpectedly, out of four our *S. aureus* strains encoding Stk2-initiated abortive infection pathway targeting staphylococcal siphoviruses, three could be productively infected with ASZ22RN. Triggering this pathway requires phage-encoded activating proteins (45). In phages phiNM1, phiNM2, and 85, this role is played by products of replication module genes directly preceding the gene for SSB protein, namely SAPPV1_GP14, AVT76_GP14, and ST85ORF023, respectively. The ASZ22RN gene preceding the SSB-encoding gene in the replication module encodes a protein (WCS65199.1) which shares 80% amino acids with SAPPV1_GP14 of phiNM1 over its entire length. Whether it is functional as the Stk2-dependent Abi pathway activator in the 1528/09 strain but cannot function in the three strains that can be productively infected with ASZ22RN, or is not functional in the activator role, remains to be found.

## MATERIALS AND METHODS

### Bacterial strains, plasmids and bacteriophages

*Staphylococcus* strains used in this study (*n* = 56) to prepare the enrichment culture, to test phage specificity, or to serve as recipient strains in transduction are listed in Table S1. Fifty-five of them were *S. aureus* strains sourced from the MICROBANK National Medicines Institute collection in Warsaw, Poland. Among these, 47 were clinical strains isolated from inpatients between 1999 and 2011, primarily from blood (*n* = 30) and wound samples (*n* = 8). Additionally, six strains were from nasal colonization in humans (*n* = 3), bovine mastitis (*n* = 2), and one from environmental samples. The isolates were characterized at both the phenotypic and molecular levels and represented 14 different genetic lineages based on the multilocus sequence typing (MLST) results. They were categorized into MLST-CCs: CC1, CC5, CC7, CC8, CC8/239, CC9, CC15, CC22, CC30, CC45, CC59, CC97, CC121, and CC398. Among the strains, 26 MRSA isolates were assigned to 19 distinct MRSA clones, which included hospital-, community-, and livestock-associated representatives from nine CCs (CC5, CC7, CC8, CC8/239, CC22, CC30, CC45, CC59, and

CC398). The remaining 29 methicillin-susceptible *S. aureus* isolates belonged to all the studied CCs. Laboratory *E. coli* strain DJ125 (76) and *S. aureus* strain RN4220, a restriction-deficient derivative of NCTC8325 cured of prophages (77, 78), were used as recipients in plasmid transformation experiments. Plasmid pMLE5 is a stably maintained *S. aureus-E. coli* shuttle expression vector that carries a chloramphenicol resistance marker for selection in *S. aureus* cells and an ampicillin resistance marker for selection in *E. coli* cells (63). Plasmid pLKA18 is a derivative of pMLE5 carrying the *terS* gene of phage ASZ22RN. It was constructed by the replacement of the pMLE5 ClaI-SphI polylinker fragment with the ASZ22RN genome fragment (coord. 1–495) obtained by the amplification of the ASZ22RN *terS* gene with primers OMLO1113 (5′-TATATATCGATTTGATTAAATTAACAC CGAAGCAAGAAAAGTTTGTATTA) and OMLO1114 (5′-TTGCATGCTTATTCATTGACGATCACT TCCGTTATTGC) and digestion with SphI and ClaI. Plasmids pLKA35 and pLKA36 are derivatives of pMLE5 containing the cloned *immR* and *immRA* genes, respectively, of phage ASZ22RN along with their preceding regions. They were constructed by replacing the ClaI-SphI polylinker fragment of pMLE5 with ASZ22RN genomic fragments corresponding to coordinates 28,889–29,375 for pLKA101 and 28,415–29,375 for pLKA102. The respective DNA fragments were amplified from ASZ22RN genomic DNA using primer pairs OMLO1683 (5′-TATTAATCGATTTGCTCACCTCCGTTTTTT) and OMLO1684 (5′-T TGCATGCTTACTTACGTTTACTTCTTATATAATCTGCATA), for pLKA101, or OMLO1683 and OMLO1685 (5′-TTGCATGCTTATATTTCCTTATATTTAAAAACTCTCAACG), for pLKA102. The PCR products were then digested with SphI and ClaI prior to ligation into the vector. The sequence correctness of all cloned inserts was confirmed by Sanger sequencing using primers OAGL41 (5′-CTCAAGGGCATCGGTCGAAC) and OAGL42 (5′-CCGGGCATTCGAAGAA TGGG).

## Bacterial growth conditions

Bacteria were grown in Luria broth (LB, Difco) or tryptic soy broth (TSB; Difco) liquid medium or in LB/TSB medium solidified with 1.5% agar (TSA; Difco) as described previously (79). LCA medium (tryptone 10 g, yeast extract 5 g, sodium chloride 10 g, $MgSO_4 \times 7H_2O$ 2.5 g, 0.25M $CaCl_2$ 10 mL per 1 L) solidified with 0.7% agar (Difco) was used for spot tests. For experiments with phage infection, LB medium was supplemented with $Ca^{2+}$ and $Mg^{2+}$ ions to the final concentration of 5 mM. When indicated, antibiotics were added to the media to the following concentrations: ampicillin 100 µg/mL, chloramphenicol 20 µg/mL.

## Phage isolation

Overnight cultures of 14 *S. aureus* strains (two laboratory strains and 12 clinical isolates; Table S1) grown in LB liquid medium at 37℃ with shaking were added in equal amounts (25 µL of each) to 45 mL of fresh liquid medium supplemented with $CaCl_2$ and $MgSO_4$. The mixed culture was incubated at 37℃ with shaking until the optical density (OD 600) of about 0.4, then incubated for 30 min without shaking and for a further 24 h with shaking at 180 RPM. Next, the culture was supplemented with 200 µL of chloroform and incubated for a further 10 min with shaking. The remaining bacteria and cell debris were harvested by centrifugation at 4℃ (9,000 RPM, MPW-380R centrifuge, rotor no. 11778). The supernatant was filtered through syringe filters of 0.45 and 0.22 µM pore diameter (Filtropour S 0.2, Sarstedt, Numbrecht, Germany). The presence of phages infective for any of the mixed culture strains in the obtained lysate was tested by a spot test. Briefly, overnight culture of each enrichment culture strain (0.1 mL) was mixed with LB medium (1 mL) and molten LCA agar (4.5 mL; cooled to ~50℃) and poured on top of LB solid medium in a Petri dish. Upon solidification, 10 µL spots of filtrate (serially diluted) were dropped on the surface of each plate and left to dry. Plates were incubated overnight at 37℃. Single plaques obtained were cut out of the agar, suspended in 300 µL of LB, vortexed, and left for 2 h to allow for phage diffusion. The phage suspensions obtained were serially diluted and used to obtain single plaques on a layer of sensitive strain cells. The whole procedure was repeated three times to purify the phage. Final phage

suspensions were titrated on layers of sensitive strain cells and used to propagate the phage. To avoid the contamination of the newly isolated phage with other undesired phages, a prophage-free, laboratory strain (RN4220) that appeared to be sensitive to infection with this phage was used for further triple phage purification through a single plaque and used as a host for phage propagation.

## Preparation of lysates of high phage titer

Lysates of high phage titer were prepared as described previously (79). Briefly, a lysate containing phages was serially diluted, and 0.1 mL of each dilution was mixed with 0.1 mL of overnight culture of the phage propagator strain. The mixture was supplemented with $CaCl_2$ and $MgSO_4$ and incubated at room temperature for 15 min. Next, 1 mL of LB broth and 5 mL of molten LCA agar (cooled to ~50°C) were added, and the mixtures were gently mixed, poured on top of LB solid medium in Petri dishes, and left to solidify. The plates were incubated overnight at 37°C. Five milliliters of LB broth were poured on plates with overlapping borders of plaques or nearly confluent lysis, which were selected for phage collection, and the plates were incubated for about 1.5 h at room temperature to allow for phage diffusion. Next, the LB broth with phages was collected, filtered through syringe filters of 0.45 and 0.22 µM pore diameter, and the filtrate was used to titrate the phage.

## Determination of phage host range with a spot test

The ability of phage to productively infect various *Staphylococcus* strains or to cause their lysis from without was tested by a spot test. The lysate containing phages for the assays was purified from bacteriocins or lysins derived from the phage propagation strain by centrifugation at >20,000 × *g* in a microcentrifuge for 3 h at 4°C and resuspension of the pellet in the original volume of fresh LB. Overnight cultures of various *S. aureus* strains in LB medium (0.1 mL) were supplemented with $CaCl_2$ and $MgSO_4$, gently mixed with 1 mL of LB and 5 mL of molten LCA (cooled to ~50°C), overlaid on LB solid medium in Petri dishes, and left to solidify. The phage suspension was serially diluted in LB to obtain $10^9$, $10^8$, $10^7$, $10^6$, and $10^5$ PFU/mL. Ten microliters of phage suspension at each dilution were spotted on the cell layer of each strain. Plates were incubated overnight at 37°C and inspected to detect lysis zones or to determine the number and morphology of plaques.

## Phage electron microscopy and virion size estimation

Lysate containing phages (~$10^9$ pfu/mL) was centrifuged in a microcentrifuge for 3 h at 14,000 RPM at 4°C, and the pellet was resuspended in 0.1 M ammonium acetate (pH 7.2). The washing was repeated two additional times under the same conditions, and the pellet was resuspended in 1/10 the original volume of the solution. A 2 µL drop of phage suspension was applied to a formvar carbon-coated copper grid (Formvar/Carbon, 300 Mesh Cu, Agar Scientific) and left for 5 min. Excess liquid was removed from the grid with a piece of Whatman filter paper. The phages were stained with 2% uranyl acetate (pH 4.0), drained of excess dye, and the grids were left for 24 h at room temperature. Grids were examined in a JEM 1400 (JEOL Co., Japan, 2008) transmission electron microscope equipped with a high-resolution digital camera (CCD MORADA, SiS-Olympus, Germany). Observations were performed at the Laboratory of Electron Microscopy at the Nencki Institute of Experimental Biology, Warsaw, Poland. Each scale bar represents 50 nm. The estimation of virion size was based on the average of measurements of at least 20 independent virions (error in neither case exceeded 10%).

## Physiological characteristics

Phage latent period and burst size were determined based on the results of one-step growth curve experiments, which were performed according to the procedure recommended by Kropinski (80). Putative lysogens were selected from among cells of growing colonies that showed up on plaques after prolonged incubation. Cells of these colonies

were purified by streak plating on fresh LB solid medium, incubated overnight at 37°C, and tested for the presence of desired phage DNA by colony PCR with primers: 5′-TTG ATTAAATTAACACCGAAGCAAGAAAAGTTTGTATTA and 5′-TTATTCATTGACGATCACTTCCGT TATTGC. Cells of the colonies that were positive in the PCR test were used to inoculate LB liquid medium, incubated overnight with shaking at 37°C, and used as indicators in a spot test with serially diluted suspensions of ASZ22RN phage to confirm that they are lysogens. The inducibility of lysogens upon induction of the SOS response was tested using mitomycin C as an SOS inducer. Overnight cultures of RN4220 lysogens containing ASZ22RN prophage were diluted 1:50 in fresh LB medium and grown with shaking (200 RPM) at 37°C to early exponential phase (OD$_{540}$ ~0.15). Next, mitomycin C was added to the cultures to the final concentration of 2 µg/mL, and the cultures were incubated with shaking (200 RPM) at 37°C for 3 h or overnight (where indicated). After incubation, 2 mL of each culture was centrifuged in a microcentrifuge (7 min 10,000 RPM), and the supernatants were filtered through a 0.2 µM syringe filter. The titer of phages in the serially diluted filtrates was determined by the spot test, using RN4220 as the indicator strain.

## Transduction

Transduction was performed as previously described (81) with some modifications. Cells of each recipient strain were grown overnight on TSA medium at 37°C. Then the bacteria were collected from the plate surface and suspended in 1 mL of TSB supplemented with CaCl$_2$. Next, 500 µL of the cell suspension was mixed with 500 µL of the bacteriophage suspension (to obtain the multiplicity of infection [MOI] ≤ 1) and 1.5 mL of TSB with CaCl$_2$ in a 15 mL test tube, and incubated for 20 min at 30°C with shaking (200 RPM). After that time, 1 mL of cold (4°C) 20 mM sodium citrate was added to the mixture to stop further phage adsorption. The mixture was centrifuged at 3,000 × *g* at 4°C for 10 min. The supernatant was discarded, and the pellet was resuspended in 1 mL of TSB containing sodium citrate (0.5 mg/mL) and incubated at 30°C for 1.5 h. The suspension was centrifuged again under similar conditions, and the pellet was resuspended in 1 mL of cold (4°C) 20 mM sodium citrate. Aliquots of the suspension (100 µL) were plated on TSA medium supplemented with sodium citrate (0.5 mg/mL) and chloramphenicol (20 µg/mL). The plates were incubated for 2 days at 30°C, and the colonies of transductants were counted. Transduction efficiency was shown in transduction units (TrU) as described by Chen et al. (21). One TrU is the number of transductants obtained from 1 mL of phage-treated cell suspension per $1 × 10^9$ PFU/mL (ratio of plasmid-carrying ASZ22RN particles to all ASZ22RN particles, where the number of plasmid-carrying particles was assumed to be equal to the number of obtained transductants per 1 mL of lysate). Each experiment was performed in triplicate.

## Isolation of plasmid DNA and DNA manipulation experiments, PCR reactions

Plasmid DNA from *E. coli* cells was isolated with the use of Plasmid Mini Kit (A&A Biotechnology, Gdańsk, Poland) according to the protocol of the kit supplier. DNA restriction digestions and ligations were performed according to enzyme supplier protocols (Thermo Fisher Scientific Baltics, Vilnius, Lithuania). Restriction fragments were separated in 1% agarose gels (82). When necessary, they were purified from a gel with the use of Gel Out Kit (A&A Biotechnology, Gdańsk, Poland). PCR reactions were performed with the use of DreamTaq DNA polymerase (Thermo Fischer Scientific Baltics, Vilnius, Lithuania) or Pfu DNA polymerase (Promega Corporation, Fitchburg, USA) according to polymerase suppliers' protocols.

## Isolation, sequencing, and sequence assembly of phage and bacterial DNA

Phages for DNA sequencing were precipitated from the DNase- and RNase-treated cell lysates of high titer ($10^{10}$ pfu/mL) using polyethylene glycol and NaCl, as described previously (75). DNA was isolated from phage particles with the use

of Qiagen Lambda Midi Kit (Qiagen Inc., Chatsworth, CA, USA) according to the supplier's instructions.

Bacteria for DNA sequencing were grown overnight in LB medium. Total bacterial DNA was isolated with the use of the Genomic Midi Kit (A&A Biotechnology, Gdańsk, Poland) according to the protocol of the kit supplier for *S. aureus* DNA isolation.

The quality control of DNA for sequencing was performed by measuring the absorbance at 260/230 nm. Template concentration was determined using a Qubit fluorometer (Thermo Fisher Scientific, Waltham, USA). DNA integrity was verified upon electrophoretic separation in 0.7% agarose gel and by capillary pulse-field gel electrophoresis (PFGE) using gDNA 165 kb kit and the FemtoPulse instrument (Agilent, Santa Clara, USA).

Illumina-compatible paired-end sequencing libraries were constructed using the NEB Ultra II FS Preparation Kit (New England Biolabs, Beverly, USA) according to the manufacturer's instructions. Libraries were sequenced using an Illumina MiSeq platform (Illumina, San Diego, USA) with 2 × 300 paired-end reads using v.3 600-cycle sequencing kit and 2 × 150 paired-end reads using v.2 300-cycle sequencing kit. Sequence quality metrics were assessed using FASTQC (http://www.bioinformatics.babraham.ac.uk/projects/fastqc/) and quality trimmed using fastp (83).

Unsheared phage DNA was taken as an input for nanopore library construction using the standard ligation sequencing kit LSK109 (Oxford Nanopore Technologies, Oxford, UK). Template DNA was end-repaired and A-tailed using NEB Ultra II End Repair Module allowing 1D adapter ligation. FFPE DNA Repair Mix was also used to repair possible nicks in the phage genome. Incubation time at this step was extended to 1 h. Sequencing was conducted on GridION X5 sequencer with FLO-MIN106 (R9.4.1) flowcells (Oxford Nanopore Technologies, Oxford, UK).

Raw nanopore data were basecalled using Guppy v.5.0.7 in super accuracy mode (Oxford Nanopore Technologies, Oxford, UK). After quality filtering using NanoFilt (84) and residual adapter removal using Porechop (https://github.com/rrwick/Porechop), the obtained data set was quality checked using NanoPlot (84). Long nanopore reads were then assembled in hybrid mode using Unicycler v.0.4.8 (85).

DNA sequence reads representing DNA of phage-transducing particles were fished out from the remaining sequence reads by comparison with the DNA sequences of phage propagation strain or its plasmids with the use of Geneious Prime (BioMatters, New Zealand). Only long nanopore reads corresponding in length to the length of phage DNA in virions were used for comparison with the sequence of phage, phage propagator strain, and its plasmid, to calculate the number of transducing phage particles compared to all phage particles.

## Sequence analysis and annotation

The phage sequences were annotated automatically using RAST at its website (https://rast.nmpdr.org/; 86, 87). The annotations were corrected and supplemented according to the results of analysis by Pharokka (88). Additionally, functions of certain gene products were predicted based on the results of InterProScan (89, 90) and HMMER (91) analysis, and on comparison of their predicted amino acid sequences or structures with sequences or structures of functionally characterized phage proteins with the use of BlastP and HHPred, respectively (92, 93). Protein length, molecular weight, and isoelectric point were determined with the use of ProtParam (94). Transmembrane domains and signal sequences were predicted using DeepTMHMM (95) and SignalP 6.0 (96), respectively. All bioinformatic analyses were performed using default program settings.

The annotations of bacterial sequences were added by the NCBI Prokaryotic Genome Annotation Pipeline (https://www.ncbi.nlm.nih.gov/genome/annotation_prok/) (97–99). The spaTyper 1.0 program (100; https://cge.food.dtu.dk/services/spaTyper/) served to determine the *spa* types of some of the sequenced genomes. A search for potential anti-phage defense loci in the genomic sequences was performed with the use of DefenseFinder at its website (https://defensefinder.mdmlab.fr/) (101–103). Additionally,

potential CRISPR-Cas loci were identified with the use of CRISPRCasFinder (104) at the Proksee server website (105).

## Phylogenetic analysis

The close relatives of newly isolated phage among phages of completely sequenced genomes from the GenBank database were found with the use of BlastN (https:// blast.ncbi.nlm.nih.gov/Blast.cgi) (92). Their taxonomic assignment was determined with the help of the International Committee for Taxonomy of Viruses database (https:// ictv.global/taxonomy). The genomic sequence identities of newly isolated phage with closely related phages were calculated with the use of the Virus Intergenomic Distance Calculator (106). The phylogenetic relatedness of a newly isolated phage to other phages was determined based on genome-wide sequence similarities calculated by tBLASTx with the use of ViPTree (https://www.genome.jp/viptree/) (107), which utilizes the proteomic tree concept developed by Rohwer and Edwards (108).

## AUTHOR AFFILIATIONS

[1]Institute of Biochemistry and Biophysics, Polish Academy of Sciences, Warsaw, Poland
[2]Doctoral School of Molecular Biology and Biological Chemistry, Institute of Biochemistry and Biophysics PAS, Warsaw, Poland
[3]Department of Epidemiology and Clinical Microbiology, National Medicines Institute, Warsaw, Poland

## AUTHOR ORCIDs

Łukasz Kałuski  http://orcid.org/0000-0003-0089-4342
Małgorzata Łobocka  http://orcid.org/0000-0003-0679-5193

## FUNDING

| Funder | Grant(s) | Author(s) |
|---|---|---|
| National Science Centre OPUS Grant | 2019/33/B/NZ2/02006 | Małgorzata Łobocka |
| Statutory Funds for the Institute of Biochemistry and Biophysics, PAS | | Małgorzata Łobocka |

## AUTHOR CONTRIBUTIONS

Łukasz Kałuski, Conceptualization, Data curation, Formal analysis, Investigation, Methodology, Validation, Visualization, Writing – original draft, Writing – review and editing | Emil Stefańczyk, Conceptualization, Data curation, Formal analysis, Investigation, Methodology, Validation, Visualization, Writing – original draft, Writing – review and editing | Aleksandra Głowacka-Rutkowska, Conceptualization, Data curation, Formal analysis, Investigation, Methodology, Validation, Writing – original draft, Writing – review and editing | Jan Gawor, Formal analysis, Investigation, Methodology, Resources, Software, Validation, Writing – original draft, Writing – review and editing | Joanna Empel, Conceptualization, Data curation, Formal analysis, Funding acquisition, Investigation, Methodology, Resources, Software, Supervision, Validation, Writing – original draft, Writing – review and editing | Monika Orczykowska-Kotyna, Investigation, Methodology, Writing – original draft | Aleksandra Szczypkowska, Investigation, Methodology, Writing – original draft | Karolina Żuchniewicz, Investigation, Methodology, Writing – original draft | Robert Gromadka, Data curation, Formal analysis, Resources, Software, Writing – original draft | Małgorzata Łobocka, Conceptualization, Data curation, Formal analysis, Funding acquisition, Investigation, Methodology, Project administration, Resources, Software, Supervision, Validation, Visualization, Writing – original draft, Writing – review and editing

## DATA AVAILABILITY

The phage ASZ22RN complete genomic sequence has been deposited in GenBank under accession number ON513432. The GenBank accession numbers of the complete *S. aureus* genomic sequences determined in the course of this work are provided in Table S1.

## ADDITIONAL FILES

The following material is available online.

### Supplemental Material

**Supplemental material (Spectrum03332-24-S0001.pdf).** Tables S1 to S11, Figures S1 to S7, and supplemental references.

### Open Peer Review

**PEER REVIEW HISTORY (review-history.pdf).** An accounting of the reviewer comments and feedback.

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
