## [Reviewer comments · Microbiology Spectrum]

Microbiology Spectrum

Characterization of a Novel *Phi*tavirus Genus Bacteriophage and Its Potential for Efficient Transfer of Modified Shuttle Plasmids to *Staphylococcus aureus* Strains of Different Clonal Complexes

Łukasz Kałuski, Emil Stefańczył, Aleksandra Głowacka-Rutkowska, Jan Gawor, Joanna Empel, Monika Orczykowska-Kotyła, Aleksandra Szczypkowska, Karolina Żuchniewicz, Robert Gromadka, and Małgorzata Łobocka

Corresponding Author(s): Małgorzata Łobocka, Institute of Biochemistry and Biophysics of the Polish Academy of Sciences

Review Timeline:

Submission Date:	December 19, 2024
Editorial Decision:	February 5, 2025
Revision Received:	April 18, 2025
Accepted:	April 23, 2025

Editor: Anne Jamet

Reviewer(s): The reviewers have opted to remain anonymous.

Transaction Report:

DOI: <https://doi.org/10.1128/spectrum.03332-24>

Re: Spectrum03332-24 (Characterization of a Novel *Phieta* Genus Bacteriophage and Its Potential for Efficient Transfer of Modified Shuttle Plasmids to *Staphylococcus aureus* Strains of Different Clonal Complexes)

Dear Prof. Małgorzata Barbara Łobocka:

Below you will find instructions from the Spectrum editorial office and the reviewer comments.

Revision Guidelines

Sincerely,
Anne Jamet
Editor
Microbiology Spectrum

Reviewer #1 (Comments for the Author):

The manuscript is interesting as it describes a new phage to be used for transduction between staphylococcal cells and because there is an extensive characterization of the phage. Furthermore they show that there is potentially a novel protein being made upon integration of the phage into the *yfkA* gene and they address the importance of lysis from without for transduction. The manuscript is well written and the only major caveat is that the discussion is extremely long and should be cut to about half the length. Many results are repeated and by deleting these repetitions, some space can be saved.

Specific comments:

- L. 384-385: The authors are surprised that a plasmid can be transduced into the strain 463/10 that harbours the prophage however this is good example of auto-transduction previous described: doi: 10.1038/ncomms13333,
L 417: Is figure 6 the right data to cite for this description? There seems to be some issues in citing figures.

Reviewer #2 (Comments for the Author):

S. aureus is an important human pathogen lacking genetic tools. Using bacteriophages, and transduction, could be of special interest, especially in the context of clinical isolates. In this manuscript, the authors present a new bacteriophage, isolated from a mixed culture of *S. aureus* strains and named ASZ22RN. This phage, even though very close phylogenetically to phage 3MRA, could represent a new species of the Phietaivirus genus. In addition, the authors show that while transduction of a plasmid by ASZ22RN is very poorly efficient, an important improvement is observed when a fragment of ASZ22RN genome (corresponding to TerS) is cloned in this plasmid. Finally, the show that this plasmid can be found in phage particles (by sequencing) as concatamers.

Here are some major comments and suggestions that I hope the authors find of interest:

1. Line 172, the authors write that bacteriophage ASZ22RN was isolated from a mixed culture of 14 *S. aureus* strains. It could be important to explain which strains (clinical isolates?). Why a mix at the first place? Why saying line 314 that the source of ASZ22RN is in fact a prophage from the 4472/08 strain ?
2. Please, show the data discussed lines 185->192
3. Lines 313->333. Please provide the alignment of attP and attB sequences (supp fig. ?)
4. Line 334->344. In order to prove yfkA disruption leads to an alternative N-terminal gene sequence that produces an active YfkA protein (and suggesting its essentiality, or at least importance, for *S. aureus*), it would be important to test the actual expression of the alternative yfkA gene and the production of an alternative YfkA protein when ASZ22RN is integrated. Deleting yfkA (if not essential) is also an important control.
5. Lines 356->358. Lysis from without is defined as "bacterial cells content liberated by a distension and destruction of the cell wall following the adsorption of phages above a certain threshold". I don't understand what makes the authors state that "ASZ22RN can penetrate thanks to its puncturing machinery" strains that cannot support productive infection (i.e. lysis from without).
6. Lines 406->412. The authors found that among the sequences obtained, some DNA molecules only contained pLKA18, as concatamers. If the main interest of ASZ22RN is its potential as a powerful genetic tool, it would be important to demonstrate that upon transduction intact plasmids composed of one genomic unit are reconstructed, and that they can be purified/sequenced.
7. Lines 419->423. Is there something special about the regions of RN422 genome transduced by generalized transduction?
8. Lines 459->463. Such statement requires to verify that when the coding sequence of the prophage repressor is deleted, immunity is abolished. More generally, the paragraph on immunity seems highly descriptive and speculative. Without the experiment proposed, this part seems more appropriate in the discussion

Minor comments

1. A reference (or quick explanation) for the terms "lysis from without" and "moron" could be important for non-experts (people working on *S. aureus* and wanting to use the phage as a genetic tool but not phage experts).
2. Providing the figure legend below the figures on the same page would improve the readability
3. Please precise the figures' panel referred to in the text (ex: lines 174, 177). Cite Fig. 2B line 201.
4. Lines 227->229. Please explain collinearity between ASZ22RN gene 4 and phage 80alpha gene 44 in a dedicated figure that could also present similarity and MuF domain.
5. Line 233, Table 2 does not provide such information
6. Line 417, Fig. 6 does not provide such information. Did the authors mean Fig. S4?
7. Line 420. "articles with similar ... albeit much lower frequencies. A word is missing.
8. Some references are missing (ex: lines 198, 211, 257, 262)
9. Line 346 and 462, ASZ22RN
10. Line 495 "virulence"
11. Line 588, "certain"

Characterization of a Novel *Phietavirus* Genus Bacteriophage and Its Potential for Efficient Transfer of Modified Shuttle Plasmids to *Staphylococcus aureus* Strains of Different Clonal Complexes

“Spectrum03332-24”

Comments to the authors:

S. aureus is an important human pathogen lacking genetic tools. Using bacteriophages, and transduction, could be of special interest, especially in the context of clinical isolates. In this manuscript, the authors present a new bacteriophage, isolated from a mixed culture of *S. aureus* strains and named ASZ22RN. This phage, even though very close phylogenetically to phage 3MRA, could represent a new species of the *Phietavirus* genus. In addition, the authors show that while transduction of a plasmid by ASZ22RN is very poorly efficient, an important improvement is observed when a fragment of ASZ22RN genome (corresponding to TerS) is cloned in this plasmid. Finally, they show that this plasmid can be found in phage particles (by sequencing) as concatamers.

Here are some major comments and suggestions that I hope the authors find of interest:

1. Line 172, the authors write that bacteriophage ASZ22RN was isolated from a mixed culture of 14 *S. aureus* strains. It could be important to explain which strains (clinical isolates?). Why a mix at the first place? Why saying line 314 that the source of ASZ22RN is in fact a prophage from the 4472/08 strain ?
2. Please, show the data discussed lines 185->192
3. Lines 313->333. Please provide the alignment of attP and attB sequences (supp fig. ?)
4. Line 334->344. In order to prove *yfkA* disruption leads to an alternative N-terminal gene sequence that produces an active YfkA protein (and suggesting its essentiality, or at least importance, for *S. aureus*), it would be important to test the actual expression of the alternative *yfkA* gene and the production of an alternative YfkA protein when ASZ22RN is integrated. Deleting *yfkA* (if not essential) is also an important control.
5. Lines 356->358. Lysis from without is defined as “bacterial cells content liberated by a distension and destruction of the cell wall following the adsorption of phages above a certain threshold”. I don't understand what makes the authors state

that “ASZ22RN can penetrate thanks to its puncturing machinery” strains that cannot support productive infection (i.e. lysis from without).

6. Lines 406->412. The authors found that among the sequences obtained, some DNA molecules only contained pLKA18, as concatamers. If the main interest of ASZ22RN is its potential as a powerful genetic tool, it would be important to demonstrate that upon transduction intact plasmids composed of one genomic unit are reconstructed, and that they can be purified/sequenced.

7. Lines 419->423. Is there something special about the regions of RN422 genome transduced by generalized transduction?

8. Lines 459->463. Such statement requires to verify that when the coding sequence of the prophage repressor is deleted, immunity is abolished. More generally, the paragraph on immunity seems highly descriptive and speculative. Without the experiment proposed, this part seems more appropriate in the discussion

Minor comments

1. A reference (or quick explanation) for the terms “lysis from without” and “moron” could be important for non-experts (people working on *S. aureus* and wanting to use the phage as a genetic tool but not phage experts).
2. Providing the figure legend below the figures on the same page would improve the readability
3. Please precise the figures' panel referred to in the text (ex: lines 174, 177). Cite Fig. 2B line 201.
4. Lines 227->229. Please explain collinearity between ASZ22RN gene 4 and phage 80alpha gene 44 in a dedicated figure that could also present similarity and MuF domain.
5. Line 233, Table 2 does not provide such information
6. Line 417, Fig. 6 does not provide such information. Did the authors mean Fig. S4?
7. Line 420. “articles with similar ... albeit much lower frequencies. A word is missing.
8. Some references are missing (ex: lines 198, 211, 257, 262)
9. Line 346 and 462, ASZ22RN
10. Line 495 “virulence”
11. Line 588, “certain”

Prof. Małgorzata Łobocka, PhD.
Head of the Laboratory of Bacteriophage Biology
Institute of Biochemistry and Biophysics
of the Polish Academy of Sciences,
Ul. Pawińskiego 5A,
02-106 Warsaw, Poland

Responses to Reviewers' Comments

Dear Editor, dear Reviewers,

We would like to thank you for all the valuable comments and suggestions. We found them very helpful in improving the quality and clarity of our manuscript. Please find below our point-by-point responses to the reviewers' comments.

Reviewers comments are in blue and authors' responses are in black.

Reviewer #1 :

The manuscript is interesting as it describes a new phage to be used for transduction between staphylococcal cells and because there is an extensive characterization of the phage. Furthermore they show that there is potentially a novel protein being made upon integration of the phage into the *yfkA* gene and they address the importance of lysis from without for transduction. The manuscript is well written and the only major caveat is that the discussion is extremely long and should be cut to about half the length. Many results are repeated and by deleting these repetitions, some space can be saved.

We thank the reviewer for all comments. We shortened the discussion section substantially, to avoid redundancies.

Specific comments:

L. 384-385: The authors are surprised that a plasmid can be transduced into the strain 463/10 that harbours the prophage however this is good example of auto-transduction previous described: doi: 10.1038/ncomms13333,

Response

We thank the reviewer for this especially helpful comment. Indeed, what we observed can be related to the phenomenon of auto-transduction. We have added the extensive explanation of our observation to the Discussion section (See L. 585-593)

“The transducibility of 463/10 strain may be related to a phenomenon originally described as auto-transduction, in which a prophage-containing strain acquires genetic material captured by transducing particles of a phage that was spontaneously induced from its own prophage and occasionally packed heterologous DNA by propagation in another susceptible strain (65). The prophage repressor of 463/10 strain protects this strain from productive infection with phage ASZ22RN by binding to the *lys*-*lysogeny* control region in the injected ASZ22RN DNA, which is similar to that of the prophage in the 463/10 strain. However, it apparently cannot prevent the entry and establishment of plasmid DNA transduced by the virions of ASZ22RN.”

L 417: Is figure 6 the right data to cite for this description? There seems to be some issues in citing figures.

Response

We apologize for the incorrect reference to Figure 6. It has been corrected in the revised manuscript and now refers to the appropriate figure, which is Figure S7.

Reviewer #2 (Comments for the Author):

S. aureus is an important human pathogen lacking genetic tools. Using bacteriophages, and transduction, could be of special interest, especially in the context of clinical isolates. In this manuscript, the authors present a new bacteriophage, isolated from a mixed culture of *S. aureus* strains and named ASZ22RN. This phage, even though very close phylogenetically to phage 3MRA, could represent a new species of the Phietavirus genus. In addition, the authors show that while transduction of a plasmid by ASZ22RN is very poorly efficient, an important improvement is observed when a fragment of ASZ22RN genome (corresponding to TerS) is cloned in this plasmid. Finally, the show that this plasmid can be found in phage particles (by sequencing) as concatamers.

Here are some major comments and suggestions that I hope the authors find of interest:

1. Line 172, the authors write that bacteriophage ASZ22RN was isolated from a mixed culture of 14 *S. aureus* strains. It could be important to explain which strains (clinical isolates?). Why a mix at the first place? Why saying line 314 that the source of ASZ22RN is in fact a prophage from the 4472/08 strain ?

Response

We revised the unclear sentence on line 172 by adding a reference to Table S1, which contains a list of all *S. aureus* strains used in our study (see L. 172-173 of the revised manuscript).

“Bacteriophage vB_SauS_ASZ22RN (ASZ22RN) was isolated from a mixed culture of 14 *S. aureus* strains (Table S1, strains marked with #) as infecting laboratory strain RN4220.”

The strains used to prepare the mixed culture (screened for the presence of a new phage) are marked in Table S1 with a “#” (Pages 1-3 of Supplemental File). These strains represent seven clonal complexes. Our aim was to isolate a new phage derived from spontaneously induced prophages within this mixed culture. The phage we isolated, infected the laboratory strain RN4220 and was designated ASZ22RN. It was subsequently studied and is described in detail in this manuscript.

A comparison of the complete genomic sequence of phage ASZ22RN with the genomes of the strains used in the mixed culture revealed its identity with one of the prophages of strain 4472/08, indicating that this prophage is the origin of phage ASZ22RN. (See L. 319-323)

“Comparison of genomic sequences of enrichment culture strains with the sequence of ASZ22RN revealed that the source of ASZ22RN phage was a prophage of 4472/08 strain, which represents the MRSA strain of clonal complex 7 (CC7) isolated from human blood. DNA sequences of the ASZ22RN phage and a prophage integrated with the genome of this strain are identical.”)

2. Please, show the data discussed lines 185->192

Response

To provide the requested data, we supplemented Figure S2 with panel B, which illustrates the differences in sensitivity between the wild-type RN4220 strain and its ASZ22RN lysogen to infection with phage ASZ22RN (See page 28 of Supplemental File).

3. Lines 313->333. Please provide the alignment of attP and attB sequences (supp fig. ?)

Response

We thank the reviewer for this comment. We have added a figure showing the sequence of the phage ASZ22RN *attP* site and its alignment with the *attB* site of phage 4472/08 to the Supplemental File as Figure S4. Additionally, to avoid repeating information illustrated in Figure S4, we have shortened the corresponding paragraph in the Results section (See page 30 of Supplemental File, and L. 323-329:

“The borders of these two sequences identity in the genome of 4472/08 strain indicated the bacterial attachment site (*attB*) for ASZ22RN prophage and the phage attachment site (*attP*) in the ASZ22RN genome (Fig. S4). The *attP* and *attB* of ASZ22RN appeared to be identical to those of staphylococcal *Dubovvirus* genus phages phi11 [NC_004615.1] and phiNM1, and to staphylococcal *Peeveelvirus* phage phiPV83-pro (Fig. S4) (10; 11; 37). The regions of two tandem direct repeats involved in the recognition by integrase is identical in ASZ22RN and phi11 [NC_004615.1].“

4. Line 334->344. In order to prove yfkA disruption leads to an alternative N-terminal gene sequence that produces an active YfkA protein (and suggesting its essentiality, or at least importance, for S. aureus), it would be important to test the actual expression of the alternative yfkA gene and the production of an alternative YfkA protein when ASZ22RN is integrated. Deleting yfkA (if not essential) is also an important control.

Response

Based on the results of our analysis, we can only predict the ability of the ASZ22RN prophage to provide a coding sequence for an alternative N-terminus of the YfkA protein. Accordingly, we have emphasized in the manuscript that this remains a prediction requiring further experimental confirmation. To support our hypothesis, we have supplemented Figure S5 with panel B (Figure S5B, see page 30 in the Supplemental File), which presents the DNA sequence of the ASZ22RN prophage upstream of the predicted start site of the proposed alternative *yfkA* variant. This region contains sequence motifs characteristic of a Shine-Dalgarno site and a transcriptional promoter recognized by the housekeeping *S. aureus* RNA polymerase. Experimental validation of these observations will require additional time and is best suited for a separate publication. However, we believe that, even at this predictive stage, these findings are valuable and may guide future research on *S. aureus* and its prophages. (See L. 337-350:

“However, we noted that the prophage terminal region proximal to the remaining part of *yfkA* gene could encode, by prediction, an alternative 16 aa, N-terminal region of YfkA that could potentially replace the original 29 aa N-terminus of this protein (Fig. S5A). Eight amino acid residues of this region are identical to those at the relevant region of *S. aureus* YfkA protein and one represents a conservative replacement. Four nucleotide residues upstream of the alternative beginning of *yfkA* gene, there is a sequence (TAAGGAGTTTATA) that resembles the typical staphylococcal Shine-Dalgarno region in mRNA and complementary to the 3' end of ribosomal 16S rRNA (underlined bases) which is involved in the initiation of translation. This region is preceded by sequences resembling -35 and -10 elements of promoters for the housekeeping RNA polymerase (Fig. S5B). Further studies will be required to con

firm, whether cells harbouring the ASZ22RN prophage, or prophages with related *attP* and the integrase binding sites, produce the alternative form of YfkA protein.”)

5. Lines 356->358. Lysis from without is defined as "bacterial cells content liberated by a distension and destruction of the cell wall following the adsorption of phages above a certain threshold". I don't understand what makes the authors state that "ASZ22RN can penetrate thanks to its puncturing machinery" strains that cannot support productive infection (i.e. lysis from without).

Response

To improve clarity, we expanded our description and included a reference to the manuscript by Kizziah et al. (2020), which describes the events following the adsorption of phage 80 α —a close relative of ASZ22RN—to *S. aureus* cells. (See L. 361-368:

“ This indicates that the observed lysis zones resulted from phage-mediated lysis from without—a phenomenon caused by phage adsorption to bacteria at a threshold level, leading to cell wall distension and destruction with subsequent release of cellular contents (Delbrück, 1940; Abedon, 2011). Clearly, the cell envelopes of all tested strains that do not support productive ASZ22RN infection are still capable of binding ASZ22RN and can be penetrated by its cell-puncturing machinery. This is due to the action of the phage tail tip complex, which has been described in detail for phage 80 α , a close relative of ASZ22RN (Kizziah et al., 2020; see also Fig. 3).”

6. Lines 406->412. The authors found that among the sequences obtained, some DNA molecules only contained pLKA18, as concatamers. If the main interest of ASZ22RN is its potential as a powerful genetic tool, it would be important to demonstrate that upon transduction intact plasmids composed of one genomic unit are reconstructed, and that they can be purified/sequenced.

Response

As requested, we confirmed that the pLKA18 plasmid transferred to RN4220 cells via ASZ22RN-mediated transduction migrates in agarose gel at the same position as the pLKA18 plasmid introduced into RN4220 cells by transformation. The corresponding agarose gel image has been added to the Supplemental File as Figure S6 (See Supplemental File, page 31). The description of the obtained results is included in the manuscript (See L. 421-425:

“To verify whether the intact plasmid was reconstructed in the chloramphenicol resistant transductants we compared, by agarose gel electrophoresis, the migration pattern of DNA molecules isolated from a transductant with the migration pattern of DNA molecules isolated from a transformant of RN4220 cells with the pLKA18 plasmid (Fig. S6). They were similar.”)

7. Lines 419->423. Is there something special about the regions of RN4220 genome transduced by generalized transduction?

Response

The pattern of RN4220 genomic region coverage, relative to the ASZ22RN attachment site, in ASZ22RN transducing particles containing fragments of RN4220 chromosomal DNA is similar to that described previously for phage phi11, which integrates with a chromosome at the same *attB* site as ASZ22RN (Bowring et al., 2022; doi: 10.1128/spectrum.02423-21). It is largely consistent with the known ability of *S. aureus* transducing phages related to ASZ22RN to mediate lateral, specialized,

and generalized transduction. Certain regions, apparently transduced by generalized transduction, are represented in the DNA of transducing particles at slightly higher levels than others. We examined whether these regions contain sequences significantly similar to those of ASZ22RN but found no such similarities. We did not investigate this further, as our observations are consistent with those reported by Bowring et al. (2022).

8. Lines 459->463. Such statement requires to verify that when the coding sequence of the prophage repressor is deleted, immunity is abolished. More generally, the paragraph on immunity seems highly descriptive and speculative. Without the experiment proposed, this part seems more appropriate in the discussion.

Response

Removal of the prophage repressor triggers prophage induction and cell lysis. As a result, it is not possible to obtain viable cells containing otherwise functional prophages that lack their repressor genes. To verify ImmR-mediated immunity to ASZ22RN, we introduced a plasmid carrying the cloned *immR* gene from ASZ22RN into RN4220 cells and demonstrated that the transformants were immune to ASZ22RN infection. However, when the plasmid also carried the *immA* gene, which encodes the predicted ImmR-inactivating protease, no immunity was observed. These results, presented in additional Figure 7, confirm the role of ImmR as a repressor of ASZ22RN lytic development. Additionally, we have included a description of this experiment at the end of the Results section. (See . L. 476-480:

“Consistently, when we introduced the cloned ASZ22RN *immR* gene to cells of the RN4220 strain, the cells of transformants acquired the immunity to infection with phage ASZ22RN (Fig. 7). When the cloned DNA fragment contained in addition to *immR*, the *immA* gene that by prediction encodes the ImmR-inactivating protease, the resistance of transformants to the infection was not observed.”)

Minor comments

1. A reference (or quick explanation) for the terms "lysis from without" and "moron" could be important for non-experts (people working on *S. aureus* and wanting to use the phage as a genetic tool but not phage experts).

Response

The terms 'lysis from without' and 'moron' were explained in the text, and relevant references supporting the definitions of each term were provided. (See L. 360-364:

“The phage used for strain sensitivity testing was purified from cell-derived lysins or bacteriocins by washing. This indicates that the observed lysis zones resulted from phage-mediated lysis from without, a phenomenon caused by phage adsorption to bacteria at a threshold limit and associated with the contents liberation by a distension and destruction of the cell wall (42, 43).”

and L. 260-262:

“Three genes next to it (33, 34,35) apparently represent a moron - an incremental addition to the phage genome, which has nothing to do with the neighbouring genes (33).”

2. Providing the figure legend below the figures on the same page would improve the readability

Response

We fully agree with this recommendation. However, according to the Microbiology Spectrum instructions for authors, figure legends must be placed immediately after the References section, and figures must be submitted separately at the end of the manuscript. To address this, we have prepared an additional file containing all figures, each accompanied by its respective legend directly below. This file has been uploaded to the submission system.

3. Please precise the figures' panel referred to in the text (ex: lines 174, 177). Cite Fig. 2B line 201.

Response

The figure panels referred to in the text were clearly indicated. The panels of Figure 1 were rearranged to match the order in which they are cited in the manuscript. Additionally, the citation of Figure 2B was added where requested.

4. Lines 227->229. Please explain collinearity between ASZ22RN gene 4 and phage 80alpha gene 44 in a dedicated figure that could also present similarity and MuF domain.

Response

As requested, the description of collinearity has been expanded. To illustrate the similarity between phage ASZ22RN Gp4 and phage 80 α Gp44, we added an alignment of the predicted MuF domain regions of both proteins, along with structural models of each protein. These additions are presented in Supplementary Fig S3 (See page 29 of Supplemental file).

5. Line 233, Table 2 does not provide such information

Response

We apologize for this oversight. The incorrect reference to Table 2 has been corrected to Table 1, which contains the cited data.

6. Line 417, Fig. 6 does not provide such information. Did the authors mean Fig. S4?

Response

We intended to refer to Fig. S4, which in the revised version of Supplementary file corresponds to Fig. S7. The incorrect reference has been corrected accordingly. (See page 32 of Supplemental file)

7. Line 420. "articles with similar ... albeit much lower frequencies. A word is missing.

Response

The entire misleading sentence has been revised for clarity (See L. 432-434:

“Other parts of the genome were also represented in the transducing particles albeit with much lower frequencies than the region to the left of ASZ22RN attachment site.”)

8. Some references are missing (ex: lines 198, 211, 257, 262)

Response

The missing references were added to the manuscript text. (See L. 199, 212, 262, 268)

9. Line 346 and 462, ASZ22RN

corrected, as requested (See L. 476, 352)

10. Line 495 "virulence"

corrected, as requested (See L. 507)

11. Line 588, "certain"

corrected, as requested (See L. 594)

Re: Spectrum03332-24R1 (Characterization of a Novel *Phieta* Genus Bacteriophage and Its Potential for Efficient Transfer of Modified Shuttle Plasmids to *Staphylococcus aureus* Strains of Different Clonal Complexes)

Dear Prof. Małgorzata Barbara Łobocka:

Your manuscript has been accepted, and I am forwarding it to the ASM production staff for publication. Your paper will first be checked to make sure all elements meet the technical requirements. ASM staff will contact you if anything needs to be revised before copyediting and production can begin. Otherwise, you will be notified when your proofs are ready to be viewed.

Sincerely,
Anne Jamet
Editor
Microbiology Spectrum